# RulePlanner: All-in-One Reinforcement Learner for Unifying Design Rules in 3D Floorplanning

**Ruizhe Zhong** [1]  **Xingbo Du** [2]  **Junchi Yan** [1]

## Abstract

Floorplanning determines the coordinate and shape of each module in Integrated Circuits. With the scaling of technology nodes, in floorplanning stage especially 3D scenarios with multiple stacked layers, it has become increasingly challenging to adhere to complex hardware design rules. Current methods are only capable of handling specific and limited design rules, while violations of other rules require manual and meticulous adjustment. This leads to labor-intensive and time-consuming post-processing for expert engineers. In this paper, we propose an all-in-one deep reinforcement learning-based approach to tackle these challenges, and design novel representations for real-world IC design rules that have not been addressed by previous approaches. Specifically, the processing of various hardware design rules is unified into a single framework with three key components: 1) novel matrix representations to model the design rules, 2) constraints on the action space to filter out invalid actions that cause rule violations, and 3) quantitative analysis of constraint satisfaction as reward signals. Experiments on public benchmarks demonstrate the effectiveness and validity of our approach. Furthermore, transferability is well demonstrated on unseen circuits. Our framework is extensible to accommodate new design rules, thus providing flexibility to address emerging challenges in future chip design. Code will be available at: https://github.com/Thinklab-SJTU/EDA-AI

This work was supported by NSFC 92370201. [1]Shanghai Jiao Tong University [2]Mohamed bin Zayed University of Artificial Intelligence. Correspondence to: Junchi Yan <yanjunchi@sjtu.edu.cn>.

*Proceedings of the $43^{rd}$ International Conference on Machine Learning*, Seoul, South Korea. PMLR 306, 2026. Copyright 2026 by the author(s).

## 1. Introduction

Floorplanning (Adya & Markov, 2003) is a critical step in modern Application-Specific Integrated Circuit design. As the very beginning of back-end physical design flow, it is recognized as an NP-hard problem (Murata et al., 1996), which defines the coordinates and shapes for all functional modules in a circuit, such as logic and memory blocks. As a prototype provided for downstream design stages, floorplanning determines the upper bound for the final Power, Performance, and Area (PPA) of the integrated circuits (Chen et al., 2024b;a).

With the scaling of technology nodes (e.g., 5-nm technology node (Sreenivasulu & Narendar, 2022)), more hardware design rules emerge in floorplanning task (Mallappa et al., 2024), especially in 3D scenarios with multiple stacked layers (Zhong et al., 2024; Cheng et al., 2005). For instance, the inter-die block alignment rule (Knechtel et al., 2015; Law et al., 2006) requires that blocks across different layers should be aligned to share a common area. Another example is the grouping constraint (Mallappa et al., 2024): a set of blocks must be physically abutted. These rules specify requirements for block position and shape, and violations can result in failure of critical functions, such as data communication and power planning.

Current methods, including analytical, heuristic-based, and reinforcement learning (RL)-based approaches, have been developed to address certain design rules, such as the non-overlap (Lu et al., 2016) and alignment (Knechtel et al., 2015) constraints. However, these methods often struggle to effectively handle multiple rules simultaneously, leading to violations of other rules. Existing legalization algorithms can only mitigate specific violations (Kai et al., 2023; Pentapati et al., 2023; Huang et al., 2024), still incapable of satisfying all design rules concurrently. Consequently, the remaining tasks for eliminating violations are left to human engineers for post-processing, which requires significant expertise and results in a time-consuming workload.

Specifically, in analytical methods (Lu et al., 2015; Lin et al., 2019; Li et al., 2022; Huang et al., 2023), the coordinates and shapes of blocks are optimized using gradient descent algorithms, which require the objective functions to

be differentiable with respect to these variables. However, the penalties for violating design constraints are often non-differentiable, limiting the effectiveness of gradient descent. Heuristic-based methods (Murata et al., 1996; Chang et al., 2000; Lin et al., 2003; Lin & Chang, 2005; Chen et al., 2007; 2014) depend on heuristic representations for floorplanning, which inadequately model the design constraints. More-over, merely including penalties for design rule violations in the cost function is insufficient for effective constraint handling. Some RL-based methods (Xu et al., 2021; Amini et al., 2022; Guan et al., 2023) utilize policy network to determine how to perturb heuristic representation, while others (Mirhoseini et al., 2021; Cheng & Yan, 2021; Lai et al., 2022; 2023; Zhong et al., 2024) directly determine the coordinates of each block, leading to a vast action space. These approaches often rely on incorporating penalties for rule violations into the reward function. However, this is insufficient for accurately addressing the complex hardware design constraints, and no additional representations are typically designed to handle these constraints. Table 1 presents a comparison of the design rules that each approach is capable of addressing. Evidently, baseline methods lack the ability to simultaneously accommodate these real-world IC design constraints.

To address these challenges, we propose RulePlanner, a deep reinforcement learning approach. Specifically, we introduce a unified framework for processing diverse hardware design rules, comprising three key components: 1) novel matrix representations for design rules, 2) action space constraints to filter invalid actions that violate these rules, and 3) quantitative metrics for design rule evaluation. Our method targets the simultaneous satisfaction of more than seven industrial design rules, addressing the complexities of real-world 3D IC design. Our main contributions are summarized as follows:

- **Unified framework for diverse design rules.** We propose RulePlanner, the first framework capable of simultaneously handling various design rules in 3D floorplanning. The framework is extensible, enabling adaptation to new design rules for future chip designs.
- **Novel representations for design rules.** We introduce new matrix representations, such as the adjacent block mask and adjacent terminal mask, to explicitly model key constraints.
- **Quantitative analysis of complex constraints.** We propose novel metrics, including block-block adjacency distance and block-terminal distance, enabling quantitative evaluation of rules.
- **Superior performance.** RulePlanner achieves effective optimization under complex design rules, outperforming previous baselines. It also demonstrates strong zero-shot generalization and transferability to unseen circuits.

*Table 1.* Comparisons of representative floorplanning approaches across various aspects, including method category (Family) and their ability to fully satisfy design rules (a-g, defined in Sec. 2.2).

| Method | Family | (a) | (b) | (c) | (d) | (e) | (f) | (g) |
|---|---|---|---|---|---|---|---|---|
| Analytical (Huang et al., 2023) | Analytical | ✗ | ✗ | ✗ | ✓ | ✗ | ✓ | ✓ |
| B*-3D-SA (Shanthi et al., 2021) | Heuristics | ✗ | ✗ | ✗ | ✗ | ✓ | ✓ | ✗ |
| WireMask-BBO (Shi et al., 2023) | Heuristics | ✗ | ✗ | ✗ | ✓ | ✓ | ✓ | ✗ |
| GraphPlace (Mirhoseini et al., 2021) | RL | ✗ | ✗ | ✗ | ✓ | ✗ | ✓ | ✗ |
| DeepPlace (Cheng & Yan, 2021) | RL | ✗ | ✗ | ✗ | ✗ | ✗ | ✓ | ✗ |
| MaskPlace (Lai et al., 2022) | RL | ✗ | ✗ | ✗ | ✓ | ✓ | ✓ | ✗ |
| FlexPlanner (Zhong et al., 2024) | RL | ✗ | ✗ | ✓ | ✓ | ✓ | ✓ | ✓ |
| RulePlanner (Ours) | RL | ✓ | ✓ | ✓ | ✓ | ✓ | ✓ | ✓ |

## 2. Preliminary and Formulation

### 2.1. 3D Floorplanning

The 3D floorplanning problem aims to determine the position and shape of each block across multiple dies or layers. Each die (layer) $d \in \mathcal{D}$ is a rectangular region with width $W$ and height $H$. The block set is denoted as $\mathcal{B} = \{b_1, b_2, \ldots, b_n\}$, where each block $b_i$ is a rectangle with width $w_i$, height $h_i$, and area $a_i = w_i \cdot h_i$, placed at coordinate $(x_i, y_i)$ on layer $z_i$. Blocks are categorized as hard (fixed aspect ratio $\mathrm{AR}_i = \frac{w_i}{h_i}$) or soft (variable aspect ratio $\mathrm{AR}_i \in [\mathrm{AR}_{\min}, \mathrm{AR}_{\max}]$ with $a_i = w_i \cdot h_i$). The terminal list is $\mathcal{T} = \{t_1, t_2, \ldots, t_m\}$, where each terminal $t_j$ has a fixed position $(x_j, y_j, z_j)$. The netlist defines connectivity among blocks and terminals. Given pre-assigned layers $z_i$ for all blocks, the objective is to optimize the coordinates $(x_i, y_i)$ and aspect ratios $\mathrm{AR}_i$ to meet specified constraints and objectives. The key challenge of 3D floorplanning lies in ***the requirement to simultaneously satisfy hardware design constraints*** (Mallappa et al., 2024; Knechtel et al., 2015), introduced in Sec. 2.2.

### 2.2. Hardware Design Rules

Design rules (constraints) for 3D floorplanning are listed as follows and demonstrated in Fig. 1. The requirement to simultaneously satisfy these rules is the critical challenge of 3D floorplanning.

(a) Boundary-constraints: Some blocks must align with a terminal located on specific edge or corner of the floorplanning outline.

(b) Grouping-constraints: Some blocks must be physically abutted, e.g., those operating on the same voltage or requiring simultaneous power-off.

(c) Inter-die block alignment constraints: Some blocks on different dies must exhibit some minimum intersecting region of their projection onto a 2D plane.

(d) Pre-placement constraints: These specify pre-defined locations and shapes of blocks.

(e) Non-overlap constraints: The overlap area between two blocks on the same die should be zero.

(f) Outline-constraint: All blocks should be placed to fit within the specified region.

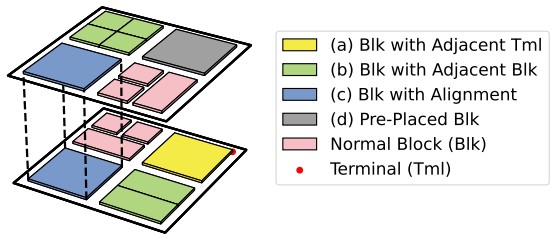

| | |
|---|---|
| 🟨 | (a) Blk with Adjacent Tml |
| 🟩 | (b) Blk with Adjacent Blk |
| 🟦 | (c) Blk with Alignment |
| ⬛ | (d) Pre-Placed Blk |
| 🟥 | Normal Block (Blk) |
| • | Terminal (Tml) |

*Figure 1.* Demonstration of hardware design rules. Specialized rules (a-d) are emphasized, while other general rules (e-g) should be satisfied by all blocks.

(g) Shape-constraints: These specify the acceptable range of width-to-height ratios of soft blocks.

## 3. Quantitative Analysis of Design Rules

To *rigorously* study design rules in Sec. 2.2, we propose the following metrics to quantitatively reflect whether the design results meet the constraint rules.

**Definition 3.1. (Block-Terminal Distance)** Given a block $b_i$ and a terminal $t_j$, we define the *block-terminal distance* between $b_i$ and $t_j$ as the minimum value among the distances between $t_j$ and four edges of $b_i$. We use the Manhattan distance to evaluate the distance between terminal $t_j$ and a linear segment $\overline{AB}$:

$$d_m(\overline{AB}, t_j) = \min_{S \in \overline{AB}} \left( |x_{t_j} - x_S| + |y_{t_j} - y_S| \right), \quad (1)$$

where $S$ is a point belonging to $\overline{AB}$. Assuming the four corner points of $b_i$ are labeled as $A, B, C, D$, the *block-terminal distance* between $b_i$ and $t_j$ is defined as follows, which should be *minimized* to satisfy the design rule (a):

$$d(b_i, t_j) = \min_{seg \in Segs(b_i)} d_m(seg, t_j). \quad (2)$$

**Definition 3.2. (Block-Block Adjacency Length)** Given two blocks $b_i, b_j$ on the same die, we define the *block-block adjacency length* $l(b_i, b_j)$ between $b_i$ and $b_j$ as follows, and it should be *maximized* to satisfy the design rule (b):

$$l_x^{ij} = \max\left(0, \min(x_i + w_i, x_j + w_j) - \max(x_i, x_j)\right).$$

$$l(b_i, b_j) = \begin{cases} l_y^{ij}, & x_i + w_i = x_j \vee x_j + w_j = x_i \\ l_x^{ij}, & y_i + h_i = y_j \vee y_j + h_j = y_i \\ 0, & \text{else}, \end{cases}$$

$$(3)$$

where $l_x^{ij}$ is the overlapping length between the linear segment starting from $x_i$ with length $w_i$ and the linear segment starting from $x_j$ with length $w_j$.

For completeness, we provide the quantitative descriptions of other constraints and objectives here. For their formal definitions, please refer to (Zhong et al., 2024; Lai et al., 2022) or Appendix B.

**Definition 3.3. (Alignment Score (Zhong et al., 2024))** Given two blocks $b_i, b_j$ on different layers, *alignment score* evaluates the overlap/intersection area between them on the common projected 2D plane. It should be *maximized* to satisfy the design rule (c).

**Definition 3.4. (HPWL (Lai et al., 2022) and Overlap (Lai et al., 2022))** Other objectives including wirelength (commonly measured with proxy Half Perimeter Wire Length, HPWL) and overlap should also be *minimized*.

## 4. Unified Framework to Tackle Design Rules

**Overview.** In RulePlanner, we frame 3D floorplanning as an episodic Markov Decision Process (MDP) solved by an actor-critic agent (Fig. 2). At each timestep $t$, the agent observes a state $s_t$, containing rule matrices and the netlist graph, and selects a hybrid action $a_t = (x, y, \text{AR})$. This action consists of a discrete position $(x, y)$ and a continuous aspect ratio AR. Design constraints are explicitly enforced on this action space. Invalid placement coordinates are filtered from the policy's output via binary masking, and the aspect ratio is clipped to satisfy shape constraints.

We introduce novel matrix representations, the adjacent terminal and block masks, to explicitly model boundary and grouping constraints. These matrices are used to generate a general mask that prunes the action space, ensuring constraint satisfaction. Our contributions also include a reward function incorporating design rule metrics and specialized neural network architectures for floorplanning. Extensibility of our approach is also discussed. To our knowledge, our method is the first to concurrently satisfy over seven complex industrial design rules.

### 4.1. Matrix Representation for Design Rule

**Adjacent Terminal Mask.** Boundary constraint specifies that a block must align with a terminal located on a specific edge or corner of the floorplanning outline. To depict this rule, we propose a new representation called adjacent terminal mask. It is a matrix $\boldsymbol{T} \in \mathbb{R}^{W \times H}$, where each element is defined as: $\boldsymbol{T}_{xy} = d(b, t)$, where $d(b, t)$ shown in Definition 3.1 is the distance between block $b$ and its adjacent terminal $t$ with block $b$ placed at $(x, y)$. Demonstration of the adjacent terminal mask is shown in Fig. 3.

Computing the adjacent terminal mask has a time complexity of $\mathcal{O}(WH)$. A naive implementation with nested loops results in low utilization of computing resources. To accelerate the calculation, we utilize the parallel capability of GPU and the efficient operator *meshgrid*. Corresponding algorithm is shown in Appendix C.2.

Under some cases, block $b_i$ has multiple adjacent terminals to align with. To tackle this requirement, we first generate the adjacent terminal mask $\boldsymbol{T}^{(ij)}$ for each block-terminal

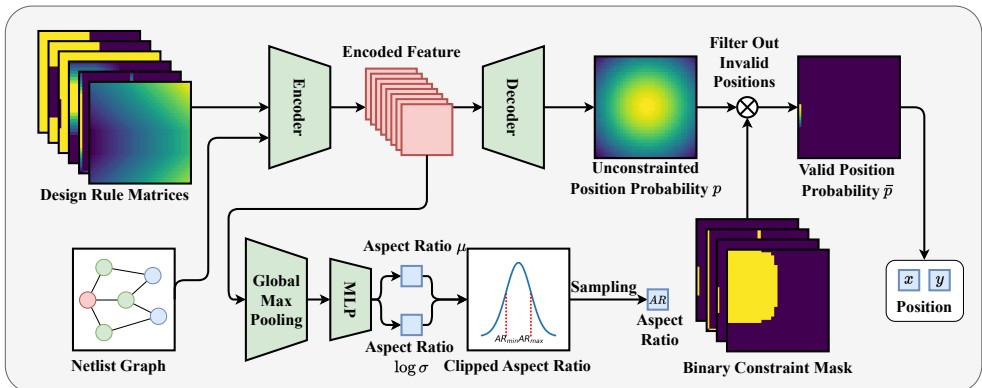

*Figure 2.* Pipeline of our approach. Stacked design rule matrices and netlist graph are taken as the input features. Hybrid action $(x, y, \text{AR})$ is determined by policy network, and is further processed to filter out invalid positions and aspect ratio range, ensuring compliance with design rules.

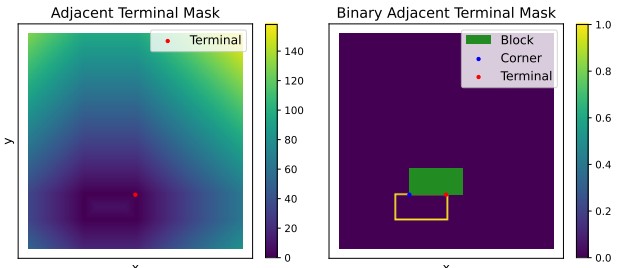

*Figure 3.* Left: the adjacent terminal mask, where each element with location $(x, y)$ indicates the distance $d(b, t)$ between terminal $t$ and block $b$ if we place $b$ at $(x, y)$. Right: the binary adjacent terminal mask, where the yellow region indicates valid positions to place block $b$ adhering to the boundary constraint. For instance, we place block $b$ at the blue point, where terminal $t$ is adjacent to it. Note that for better visualization, we move the terminal into the floorplan region instead of along the boundary.

pair, where $t_j$ is one of the adjacent terminals of block $b_i$. If $b_i$ is required to align with these terminals simultaneously, we use the operation $\max$ to merge these matrices:

$$\boldsymbol{T}_{xy}^{(i)} = \max_j \boldsymbol{T}_{xy}^{(ij)}, \forall t_j \in \mathcal{T}(b_i), \tag{4}$$

where $\mathcal{T}(b_i)$ is the set of adjacent terminals of block $b_i$. If the block is only required to align with at least one of these terminals, we can instead use the operator $\min$.

**Adjacent Block Mask.** Grouping constraint specifies a set of blocks that must be physically abutted. We propose adjacent block mask to model this rule. Given two blocks $b_i$ and $b_j$, $b_i$ is a movable block to place and $b_j$ is a placed block. The adjacent block mask between $b_i, b_j$ is denoted as $\boldsymbol{B} \in \mathbb{R}^{W \times H}$, and each element $\boldsymbol{B}_{xy} = l(b_i, b_j)$, where $l(b_i, b_j)$ shown in Definition 3.2 is the block-block adjacency length between block $b_i, b_j$ if $b_i$ is placed at location $(x, y)$. Fig. 4 demonstrates this matrix.

In certain cases, multiple (more than two) blocks should be grouped together to form a voltage island (Lin et al.,

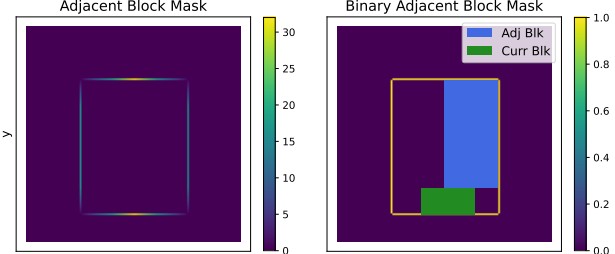

*Figure 4.* Left: The adjacent block mask for block $b_i$ (green), where each element $(x, y)$ represents the adjacency length $l(b_i, b_j)$ between $b_i$ and its placed adjacent block $b_j$ (blue). Right: The binary adjacent block mask, with the yellow region denoting valid positions for placing $b_i$ such that it is physically abutted to its adjacent block $b_j$, thereby satisfying the grouping constraint.

2018), which is a set of blocks occupying a contiguous physical space and operating at one supply voltage. To meet this requirement, we apply the operator $\text{sum}$ to merge these masks. Specifically, given a block $b_i$ and other blocks within the same voltage island, we construct the adjacent block mask for $b_i$ as follows:

$$\boldsymbol{B}_{xy}^{(i)} = \sum_j \boldsymbol{B}_{xy}^{(ij)}, \forall b_j \in \mathcal{V}(b_i), \tag{5}$$

where $\mathcal{V}(b_i)$ denotes the set of other blocks within the voltage island of $b_i$, and $\boldsymbol{B}_{xy}^{(ij)} = l(b_i, b_j)$. We further employ parallel operators to expedite matrix construction, as detailed in Appendix C.3.

### 4.2. Constraint on the Action Space

Conventional floorplanning methods typically convert constraints into penalties (Lin et al., 2019) or integrate them into the reward function (Cheng & Yan, 2021), effectively relaxing hard constraints into soft ones for optimization. However, this approach often results in longer training times and may fail to satisfy all constraints, necessitating additional

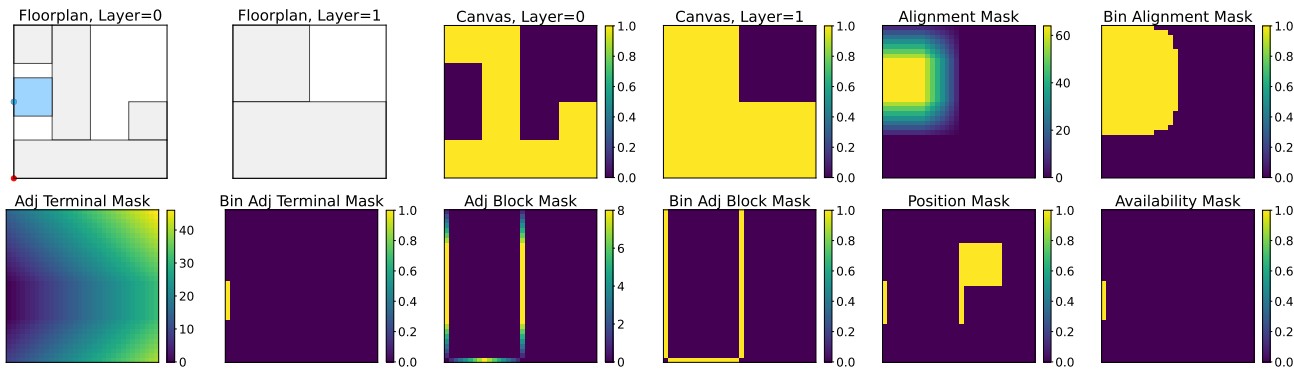

*Figure 5.* Demonstration of chip floorplan layout and all masks for the availability mask. In sub-figures 'Floorplan, Layer=0/1', current block to place is highlighted with blue, while gray blocks have already been placed. Red points represent the normal terminals, and the blue points indicate the terminals with which specific blocks should align. Adj: adjacent. Bin: binary.

post-processing steps to resolve violations such as overlaps (Kai et al., 2023; Spindler et al., 2008). For complex constraints, effective legalization algorithms are lacking, and simply incorporating penalties into the reward is often insufficient to ensure constraint satisfaction.

To address these issues, we propose a robust approach of applying direct constraints on the action space. We will discuss how to restrict the action space, including the determination of 1) block position and 2) aspect ratio, to satisfy corresponding constraints.

**Block Position.** The policy network, parameterized by $\theta$, determines the placement position for each block by generating logits $p_\theta \in \mathbb{R}^{W \times H}$. Applying the softmax operator yields a probability distribution for sampling block coordinates. To enforce floorplanning constraints, this distribution is modified using a mask. Inspired by MaskPlace (Lai et al., 2022), we propose a more general mask: the availability mask $M$. Various matrices are used to construct $M$: (1) adjacent terminal mask $T$, (2) adjacent block mask $B$, (3) inter-die block alignment mask $A$, and (4) position mask $P$. The inter-die block alignment mask (Zhong et al., 2024) $A \in \mathbb{R}^{W \times H}$ assigns an alignment score to each position $(x, y)$. The position mask $P \in \{0, 1\}^{W \times H}$ indicates valid placements that avoid overlap and out-of-bounds conditions. All masks are binarized with thresholds $\bar{t}, \bar{b}, \bar{a}$ as follows:

$$\bar{T}_{xy} = \begin{cases} 1, & T_{xy} \leq \bar{t}, \\ 0, & T_{xy} > \bar{t}, \end{cases} \qquad \bar{B}_{xy} = \begin{cases} 1, & B_{xy} \geq \bar{b}, \\ 0, & B_{xy} < \bar{b}, \end{cases}$$

$$\bar{A}_{xy} = \begin{cases} 1, & A_{xy} \geq \bar{a}, \\ 0, & A_{xy} < \bar{a}. \end{cases} \qquad \bar{P} = P.$$

$$\text{(6)}$$

The binary masks are combined to form the availability mask $M$, which is applied to the policy network's action space to filter out invalid positions:

$$M = \bar{T} \odot \bar{B} \odot \bar{A} \odot \bar{P}, \bar{p}_\theta = \text{softmax}\left(p_\theta + (M - 1) \odot 10^8\right),$$

$$\text{(7)}$$

where $\odot$ denotes element-wise multiplication. This approach eliminates infeasible placements, ensuring constraint satisfaction and reducing the action space to accelerate training.

**Block Aspect Ratio.** The aspect ratio of each soft block is adaptively selected by the policy network at each step to fit the fixed outline. To enforce shape constraints, $z$ is modeled as a Gaussian random variable with mean $\mu_\theta$ and standard deviation $\sigma_\theta$: $z \sim \mathcal{N}(\mu_\theta, \sigma_\theta^2)$. The network outputs $\mu_\theta$ using a tanh (Dubey et al., 2022) activation in the final layer, yielding $\mu_\theta \in [-1, 1]$. Final aspect ratio is projected to the valid range $[\text{AR}_{\min}, \text{AR}_{\max}]$ via affine transformation and clipping:

$$\bar{z} = \frac{(z + 1)}{2} \cdot (\text{AR}_{\max} - \text{AR}_{\min}) + \text{AR}_{\min},$$
$$\text{AR} = \text{clip}\left(\bar{z}, \text{AR}_{\min}, \text{AR}_{\max}\right).$$
$$\text{(8)}$$

### 4.3. Quantified Penalty of Rule Violation

Apart from direct constraints on the action space to filter out invalid actions, we also incorporate corresponding objectives into the calculation of reward. Specifically, block-block adjacency length, block-terminal distance, alignment score, HPWL, and overlap penalties are involved in the reward computation. Moreover, a self-adaptive and robust reward reshaping technique is utilized to achieve normalization and stabilize the training process. Details of reward calculation are shown in Sec. C.1.

### 4.4. Model Architecture for Multi-Modality Processing

We employ a Transformer-based model to extract features from the netlist graph, which is defined by a node feature matrix $X \in \mathbb{R}^{L \times d}$ and an adjacency matrix $E \in \{0, 1\}^{L \times L}$. Each node $j$ is initially represented by a feature vector $g_j = [x_j, y_j, z_j, w_j, h_j, a_j, p_j]$, comprising its 3D coordinates, dimensions, area, and a binary placement indicator. These are concatenated with features $y_v$ from a vision encoder to form the rows of the input matrix $X$.

To incorporate the graph topology, the Transformer's self-attention mechanism is constrained by an attention mask derived from $1 - \boldsymbol{E}$. Furthermore, the placement order $\boldsymbol{o} \in \mathbb{Z}^L$ is injected into the model by adding sinusoidal position encodings (Vaswani et al., 2017) to the input features. From the model's output embeddings, we derive the local feature for a target node idx and the global graph feature:

$$
\begin{aligned}
\boldsymbol{H}_1 &= \mathrm{FC}(\boldsymbol{G}) + \mathrm{PositionEncoding}(\boldsymbol{o}) \in \mathbb{R}^{L \times d} \\
\boldsymbol{H}_2 &= \mathrm{TransformerEncoder}(\boldsymbol{H}_1, \mathrm{mask} = 1 - \boldsymbol{E}) \\
\boldsymbol{e}_{local} &= \mathrm{FC}(\boldsymbol{H}_2[\mathrm{idx}]), \quad \boldsymbol{e}_{global} = \mathrm{FC}\left(\mathrm{AvgPool}(\boldsymbol{H}_2)\right) \\
\boldsymbol{y} &= \mathrm{FC}(\mathrm{concat}(\boldsymbol{y}_v, \boldsymbol{e}_{global}, \boldsymbol{e}_{local}))
\end{aligned}
\tag{9}
$$

### 4.5. Full Pipeline under RL with Hybrid Action Space

**State Space.** The state space consists of rule matrices and graph. The former includes the adjacent block mask, adjacent terminal mask, canvas image, wire mask (Lai et al., 2022), position mask, alignment mask (Zhong et al., 2024), which are concatenated along the channel dimension and processed by a CNN. The netlist graph $(\boldsymbol{G}, \boldsymbol{E})$ is encoded using our graph model.

**Hybrid Action Space.** The hybrid action space is defined as $\mathcal{X} \times \mathcal{Y} \times \mathcal{R}$, where $\mathcal{X} = \{1, 2, \dots, W\}$ and $\mathcal{Y} = \{1, 2, \dots, H\}$ are discrete sets representing spatial coordinates, and $\mathcal{R} = \mathbb{R}$ is a continuous set for aspect ratios. At each step $t$, the policy outputs: (1) a discrete probability distribution over 2D positions $(x_t, y_t) \in \mathcal{X} \times \mathcal{Y}$ for the current block $b_t$; (2) the mean $\mu_\theta$ and standard deviation $\sigma_\theta$ of a Gaussian distribution for sampling the aspect ratio $\mathrm{AR}_{t+1} \in \mathcal{R}$ of the next block $b_{t+1}$. The action $a_t$ includes the aspect ratio for $b_{t+1}$ (rather than $b_t$) because the next state $s_{t+1}$ requires the shape $(w_{t+1}, h_{t+1})$ to compute all relevant masks. To enforce constraints, the availability mask $\boldsymbol{M}$ is applied during position selection, restricting valid positions to those with $\boldsymbol{M}_{xy} = 1$. Additionally, aspect ratios are clipped to remain within the specified bounds.

**Training Pipeline.** We employ the Actor-Critic framework (Konda & Tsitsiklis, 1999) with the Hybrid Proximal Policy Optimization (Schulman et al., 2017; Fan et al., 2019) algorithm, a standard approach in related work (Mirhoseini et al., 2021; Lai et al., 2022; Zhong et al., 2024). The objective of the policy $\pi_\theta(a_t | s_t)$ is:

$$
\begin{aligned}
\bar{r}_t^{(k)} &= \mathrm{clip}\left(r_t^{(k)}, 1 - \varepsilon, 1 + \varepsilon\right), \\
L(\theta) &= \sum_{k=1}^{2} \lambda_k \cdot \hat{\mathbb{E}}_t \left[\min\left(r_t^{(k)} \hat{A}_t, \bar{r}_t^{(k)} \hat{A}_t\right)\right],
\end{aligned}
\tag{10}
$$

where $k = 1, 2$ represents position and aspect ratio decision. $\lambda_k$ is the weight for each clip loss. $r_t^{(k)} = \frac{\pi_\theta(a_t^{(k)}|s_t)}{\pi_{\theta_{old}}(a_t^{(k)}|s_t)} \cdot \hat{A}_t$

*Table 2.* Hardware design rules for each task.

| Task | Hardware Design Rules | | | | | | |
|---|---|---|---|---|---|---|---|
| | (a) | (b) | (c) | (d) | (e) | (f) | (g) |
| 1 | ✓ | | ✓ | | ✓ | ✓ | ✓ |
| 2 | | ✓ | ✓ | | ✓ | ✓ | ✓ |
| 3 | ✓ | ✓ | ✓ | ✓ | ✓ | ✓ | ✓ |

denotes the generalized advantage estimation (GAE) (Schulman et al., 2015), and $G_t = \hat{A}_t + V_t$ is the cumulative discounted reward (Schulman et al., 2015; Weng et al., 2022). We employ $\hat{A}_t = \sum_{i=0}^{T-t-1}(\gamma\lambda)^i \delta_{t+i}$, where $\delta_t = r_t + \gamma V_{t+1} - V_t$, and $G_t = \hat{A}_t + V_t$ as the cumulative discounted reward (Schulman et al., 2015; Weng et al., 2022). $V_t$ is the estimated state value from critic network $V_\phi(s_t)$, and the critic is updated with minimizing:

$$
L(\phi) = \lambda_\phi \cdot \hat{\mathbb{E}}_t \left[(G_t - V_\phi(s_t))^2\right].
\tag{11}
$$

### 4.6. Discussion about Extensibility

Our framework unifies design rule handling via a modular and extensible system. Each rule is defined by a three-part construct: (1) a **rule matrix** encoding expert knowledge into spatial preferences; (2) a derived **action space constraint** to prune invalid placements; and (3) a **quantitative metric** to penalize residual violations in the reward signal. This architecture allows new rules to be readily incorporated by defining their corresponding components, a flexibility demonstrated for future design challenges in Appendix E.

## 5. Experiment and Analysis

### 5.1. Evaluation Protocol

**Benchmarks.** We evaluate the performance of Rule-Planner on public benchmarks **MCNC** (MCNC) and **GSRC** (GSRC). They contain eight circuits with the number of blocks ranging from 10 to 300. Note that the scale of the largest circuit in them is significantly larger than the ones of most industrial circuits, as stated in FloorSet (Mallappa et al., 2024).

**Tasks.** We define tasks with distinct design constraints in Table 2. Rules (c) inter-die block alignment, (e) non-overlap, (f) outline constraint, and (g) shape constraint are treated as common requirements for all tasks, while (a) boundary constraint, (b) grouping constraint, and (d) pre-placement are considered specialized. These tasks assess the adaptability of our method to diverse constraint scenarios.

**Baselines.** Typical methods including analytical (Huang et al., 2023), heuristic-based (B*-3D-SA (Shanthi et al., 2021), WireMask-BBO (Shi et al., 2023)) and learning-based approaches (GraphPlace (Mirhoseini et al., 2021), DeepPlace (Cheng & Yan, 2021), MaskPlace (Lai et al.,

2022), FlexPlanner (Zhong et al., 2024)) are selected as baselines. Each experiment is conducted five times with different seeds. Due to the page limitation, additional details regarding the baseline comparisons and implementation specifics are provided in Appendix G, and I.

## 5.2. Main Results

We evaluate block-terminal distance in Task 1 (Table 3) and block-block adjacency length in Task 2 (Table 4). Our method achieves a **block-terminal distance** of 0.000, indicating complete alignment with terminals and full satisfaction of boundary constraints, whereas all baselines exhibit higher distances. For **block-block adjacency**, our approach attains the longest adjacency length (0.223), surpassing the best baseline (0.082) and demonstrating superior grouping constraint satisfaction. Due to page limitation, comprehensive evaluation results including **inter-die alignment, HPWL, overlap** across different tasks are presented in Appendix L.

The comparative results highlight a fundamental trade-off in the floorplanning process. Baselines like Analytical and WireMask-BBO are primarily optimized for traditional metrics such as HPWL, but this narrow focus leads to their inability to address the multifaceted design constraints of modern circuits. Our methodology represents a strategic shift in priority: we posit that ensuring full compliance with complex design rules is more critical than achieving the absolute lowest HPWL. Consequently, while our method may not outperform all baselines on HPWL across every circuit (e.g., n30 - n300), its demonstrated ability to completely satisfy intricate constraints, a task where others fail significantly, confirms the validity and necessity of this strategic trade-off.

## 5.3. Zero-shot and Fine-tune

We demonstrate our model's generalization and transferability via zero-shot and fine-tuning experiments, using a model pre-trained on the n100 circuit.

First, for zero-shot performance, the model is evaluated on other circuits without any fine-tuning. The high inference-to-training performance ratios in Table 5 indicate robust generalization across circuits of varying scales. Second, for transferability, fine-tuning from pre-trained weights achieves comparable or superior performance to training from scratch but with significantly reduced computational cost. Full results for these experiments are in Appendices M and N.

We attribute these strong transferability results to our framework's generalized constraint representation. For instance, the *adjacent block mask* guides the policy to identify optimal regions that maximize block-block adjacency, while the *availability mask* encodes expert priors to compel the agent

to adhere to design rules.

## 5.4. Ablation Studies

We conduct an ablation study to evaluate the effectiveness of the adjacent terminal mask and adjacent block mask. For each mask type, we remove it from both the input features and the action space constraints. Table 6 reports their impact on block-block adjacency length and block-terminal distance, where 'Feat.' denotes inclusion as an input feature and 'Constr.' indicates use as an action space constraint. As input features, these masks are essential for capturing design rule information. As constraints, they reduce the action space and accelerate training. The results also show that relying solely on reward-based optimization is insufficient to achieve satisfactory design quality.

## 6. Related Works

**Classical floorplanning approaches** are broadly categorized into heuristic and analytical methods. The former encode floorplans using specialized representations like B\*-trees (Chang et al., 2000; Shanthi et al., 2021), Corner Block Lists (Lin et al., 2003; Knechtel et al., 2017), or Sequence Pairs (Murata et al., 1996; Prakash & Lal, 2021). A search algorithm, most notably Simulated Annealing (Bertsimas & Tsitsiklis, 1993), then explores the solution space of these representations, which are finally decoded into a physical layout.

Analytical approaches (Li et al., 2022; Huang et al., 2023) often formulate floorplanning by analogy to an electrostatic system (Lu et al., 2015; Lin et al., 2019). These methods directly optimize block coordinates by computing the gradients of objective functions and employing gradient-based solvers.

**RL-based floorplanning approaches** have recently emerged as a promising paradigm. One line of research augments traditional, perturbation-based methods with RL, where the policy network either accepts/rejects state perturbations (Xu et al., 2021; Guan et al., 2023) or determines the perturbations directly (Amini et al., 2022). Another line of work focuses on using RL for direct block placement. These methods leverage graph or vision-based representations (Mirhoseini et al., 2021; Cheng & Yan, 2021), or employ specialized matrix representations. For instance, MaskPlace (Lai et al., 2022) introduces a wire mask to model HPWL, FlexPlanner (Zhong et al., 2024) proposes an alignment mask for inter-die constraints, and EXPlace (Gao et al., 2026) injects expert placement knowledge into RL-based macro placement; these works also use masks or dense guidance signals to steer placement decisions.

However, a critical limitation of prior works is their narrow scope. Each method is tailored to a small, specific subset of

*Table 3.* **Block-Terminal Distance** comparison on Task 1 among baselines and our method. The lower the Block-Terminal Distance, the better, and the optimal results are shown in **bold**. C/M means Circuit/Method.

| C/M | Analytical | B*-3D-SA | GraphPlace | DeepPlace | MaskPlace | WireMask-BBO | FlexPlanner | RulePlanner |
|---|---|---|---|---|---|---|---|---|
| ami33 | 0.079±0.058 | 0.237±0.078 | 0.314±0.006 | 0.333±0.044 | 0.234±0.010 | 0.609±0.000 | 0.208±0.012 | **0.000±0.000** |
| ami49 | 0.101±0.067 | 0.259±0.037 | 0.730±0.037 | 0.509±0.120 | 0.507±0.074 | 0.677±0.078 | 0.386±0.016 | **0.000±0.000** |
| n10 | 0.237±0.172 | 0.254±0.225 | 0.008±0.000 | 0.377±0.014 | **0.000±0.000** | 0.857±0.003 | 0.471±0.002 | **0.000±0.000** |
| n30 | 0.195±0.092 | 0.201±0.051 | 0.644±0.141 | 0.637±0.116 | 0.433±0.002 | 0.691±0.001 | 0.205±0.005 | **0.000±0.000** |
| n50 | 0.230±0.008 | 0.182±0.151 | 0.645±0.018 | 0.655±0.185 | 0.389±0.002 | 0.549±0.021 | 0.140±0.002 | **0.000±0.000** |
| n100 | 0.169±0.016 | 0.287±0.069 | 0.814±0.030 | 0.737±0.070 | 0.352±0.052 | 0.826±0.002 | 0.466±0.004 | **0.000±0.000** |
| n200 | 0.188±0.012 | 0.606±0.141 | 0.838±0.121 | 0.610±0.000 | 0.337±0.020 | 0.694±0.003 | 0.329±0.021 | **0.000±0.000** |
| n300 | 0.150±0.023 | 0.789±0.090 | 0.800±0.048 | 0.645±0.037 | 0.573±0.106 | 0.675±0.011 | 0.242±0.017 | **0.000±0.000** |
| Avg. | 0.169 | 0.352 | 0.599 | 0.563 | 0.353 | 0.697 | 0.306 | **0.000** |

*Table 4.* **Block-Block Adjacency Length** comparison on Task 2 among baselines and our method. The higher the Block-Block Adjacency Length, the better, and the optimal results are shown in **bold**. C/M means Circuit/Method.

| C/M | Analytical | B*-3D-SA | GraphPlace | DeepPlace | MaskPlace | WireMask-BBO | FlexPlanner | RulePlanner |
|---|---|---|---|---|---|---|---|---|
| ami33 | 0.000±0.000 | 0.030±0.026 | 0.059±0.040 | 0.056±0.014 | 0.109±0.015 | 0.118±0.003 | 0.060±0.021 | **0.251±0.001** |
| ami49 | 0.000±0.000 | 0.006±0.011 | 0.026±0.015 | 0.025±0.025 | 0.011±0.011 | 0.047±0.011 | 0.059±0.009 | **0.204±0.002** |
| n10 | 0.000±0.000 | 0.207±0.180 | 0.174±0.174 | 0.350±0.002 | 0.352±0.000 | 0.350±0.003 | 0.225±0.014 | **0.436±0.002** |
| n30 | 0.000±0.000 | 0.058±0.043 | 0.000±0.000 | 0.042±0.003 | 0.028±0.008 | 0.120±0.000 | 0.105±0.023 | **0.262±0.003** |
| n50 | 0.020±0.017 | 0.021±0.036 | 0.000±0.000 | 0.049±0.002 | 0.010±0.010 | 0.008±0.011 | 0.084±0.002 | **0.243±0.002** |
| n100 | 0.004±0.007 | 0.007±0.009 | 0.014±0.004 | 0.009±0.009 | 0.016±0.006 | 0.012±0.000 | 0.024±0.003 | **0.171±0.002** |
| n200 | 0.000±0.000 | 0.003±0.006 | 0.007±0.005 | 0.011±0.000 | 0.027±0.000 | 0.004±0.005 | 0.012±0.007 | **0.113±0.002** |
| n300 | 0.000±0.000 | 0.000±0.000 | 0.007±0.003 | 0.002±0.002 | 0.011±0.000 | 0.000±0.000 | 0.000±0.000 | **0.106±0.002** |
| Avg. | 0.003 | 0.042 | 0.036 | 0.068 | 0.070 | 0.082 | 0.071 | **0.223** |

*Table 5.* Zero-shot evaluation on Task 2. Ratio: metric between zero-shot inference and training.

| Metric/Circuit | | ami33 | ami49 | n10 | n30 | n50 | n200 | n300 |
|---|---|---|---|---|---|---|---|---|
| Blk-Blk Adj. Length (↑) | value | 0.266 | 0.214 | 0.434 | 0.277 | 0.225 | 0.128 | 0.113 |
| | ratio | 1.060 | 1.049 | 0.995 | 1.057 | 0.926 | 1.133 | 1.066 |
| Alignment (↑) | value | 0.926 | 0.913 | 0.940 | 0.952 | 0.943 | 0.943 | 0.936 |
| | ratio | 1.019 | 1.003 | 0.985 | 1.017 | 0.974 | 1.004 | 1.023 |
| HPWL (↓) | value | 66,831 | 856,516 | 32,596 | 90,679 | 116,090 | 349,203 | 481,606 |
| | ratio | 1.029 | 1.068 | 1.001 | 1.015 | 1.019 | 1.043 | 0.976 |

*Table 6.* Ablation study about the rule masks on the block-block adjacency length (↑) and block-terminal adjacency distance (↓).

| Adjacent Block Mask | | | Adjacent Terminal Mask | | |
|---|---|---|---|---|---|
| Feat. | Constr. | Length | Feat. | Constr. | Distance |
| | | 0.057 | | | 0.145 |
| ✓ | | 0.153 | ✓ | | 0.058 |
| | ✓ | 0.143 | | ✓ | 0.045 |
| ✓ | ✓ | **0.169** | ✓ | ✓ | **0.000** |

design rules, such as overlap prevention (Lai et al., 2022) or block alignment (Zhong et al., 2024), neglecting the multifaceted constraints of real-world 3D IC design.

**ML for broader chip design** has also been explored beyond floorplanning. Recent studies apply learning to logic optimization and synthesis (Bai et al., 2025; 2026), and to PPA- or cross-stage-aware macro placement (Geng et al., 2025). These works highlight the increasing importance of aligning learning-based EDA methods with practical downstream design objectives.

# 7. Conclusion and Outlook

We propose RulePlanner, an RL framework for 3D floorplanning subject to complex, real-world hardware constraints. To this end, RulePlanner introduces a novel and extensible mechanism capable of concurrently satisfying over seven industrial design rules. This is achieved by unifying them through three core components: 1) unified framework for diverse design rules, 2) new representations for design rules, and 3) quantitative violation metrics. A primary limitation, and an avenue for future work, is the absence of thermal optimization (Duan et al., 2024). We plan to address this by: 1) incorporating power maps (Park et al., 2009) into the state representation, 2) deriving the thermal profile via fast analysis (Zhan & Sapatnekar, 2005b), and 3) integrating temperature into the reward function.

# Impact Statement

This paper presents work whose goal is to advance the field of Machine Learning. There are many potential societal consequences of our work, none which we feel must be specifically highlighted here.

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

## A. Notation

In this section, we list the meaning of all notations shown in this paper, which is demonstrated in Table 7.

*Table 7.* Notation.

| Notation | Meaning |
|---|---|
| $d$ | the die/layer, the rectangular floorplanning region |
| $\mathcal{D}$ | the set of all dies/layers |
| $W$ | the width of each die |
| $H$ | the height of each die |
| $b$ | the block |
| $\mathcal{B}$ | the set of all blocks |
| $w$ | the width of a block |
| $h$ | the height of a block |
| $a$ | the area of a block |
| $x$ | the x-coordinate of a block |
| $y$ | the y-coordinate of a block |
| $z$ | the z-coordinate (die/layer index) of a block |
| AR | the aspect ratio of a block |
| $\text{AR}_{\min}$ | the lower bound of the acceptable range of aspect ratio |
| $\text{AR}_{\max}$ | the upper bound of the acceptable range of aspect ratio |
| $t$ | the terminal |
| $\mathcal{T}$ | the set of all terminals |
| $\text{aln}(i, j)$ | the alignment score between block $b_i$ and its alignment partner $b_j$ |
| $l(b_i, b_j)$ | the block-block adjacent length between block $b_i$ and its adjacent block $b_j$ |
| $d(b_i, t_j)$ | the block-terminal distance between block $b_i$ and its adjacent terminal $t_j$ |
| $\boldsymbol{T}^{(ij)}$ | the adjacent terminal mask between block $b_i$ and its adjacent terminal $t_j$ |
| $\boldsymbol{B}^{(ij)}$ | the adjacent block mask between block $b_i$ and its adjacent block $b_j$ |
| $\boldsymbol{A}^{(ij)}$ | the alignment mask between block $b_i$ and its alignment partner $b_j$ |
| $\boldsymbol{P}$ | the position mask for a block |
| $\boldsymbol{M}$ | the availability mask for a block |
| $\bar{t}$ | threshold for adjacent terminal mask |
| $\bar{b}$ | threshold for adjacent block mask |
| $\bar{a}$ | threshold for alignment mask |

## B. Definition of Alignment Score, HPWL and Overlap

**Definition B.1. (Alignment Score (Zhong et al., 2024))** Given two blocks $b_i, b_j$ on different layers, *alignment score* evaluates the overlap/intersection area between them on the common projected 2D plane. We define the alignment score $\text{aln}(i, j)$, and it should be ***maximized*** to satisfy the design rule (c):

$$
\begin{aligned}
\text{aln}_x(i, j) &= \max\left(0, \min(x_i + w_i, x_j + w_j) - \max(x_i, x_j)\right), \\
\text{aln}_y(i, j) &= \max\left(0, \min(y_i + h_i, y_j + h_j) - \max(y_i, y_j)\right), \\
\text{aln}(i, j) &= \min\left(1, \frac{\text{aln}_x(i, j) \cdot \text{aln}_y(i, j)}{\text{aln}_m(i, j)}\right),
\end{aligned}
\tag{12}
$$

where $\text{aln}_m(i, j)$ is the required minimum alignment area between block $b_i$ and $b_j$.

**Definition B.2. (HPWL)** Half Perimeter Wire Length is an approximate metric of wirelength. It can be computed much more efficiently, as accurate wirelength can be accessed only after the time-consuming routing stage. The summation of HPWL should be ***minimized***:

$$
\sum_{\text{net} \in \text{netlist}} \left( \max_{m_i \in \text{net}} x_i^c - \min_{m_i \in \text{net}} x_i^c + \max_{m_i \in \text{net}} y_i^c - \min_{m_i \in \text{net}} y_i^c \right),
\tag{13}
$$

where $m_i$ is either a block or terminal in net and $x_i^c$ is the center x-coordinate. For a block, $x_i^c = x_i + \frac{w_i}{2}$, and for a terminal, $x_i^c = x_i$.

**Definition B.3. (Overlap)** Given two blocks $b_i, b_j$ on the same die, the overlap area between them is defined as follows, and it should be *minimized* to satisfy the design rule (e):

$$o_{ij}^x = \max\left(0, \min(x_i + w_i, x_j + w_j) - \max(x_i, x_j)\right),$$
$$o_{ij}^y = \max\left(0, \min(y_i + h_i, y_j + h_j) - \max(y_i, y_j)\right),$$
$$o_{ij} = o_{ij}^x \cdot o_{ij}^y.$$

(14)

## C. Algorithm

### C.1. Reward Function Design

**Reward shaping.** Given a circuit, different metrics often exhibit vastly different magnitudes. For example, the HPWL metric is typically on the order of $10^5$, whereas the alignment score falls within the $[0, 1]$ interval. Since the reward function needs to aggregate these heterogeneous signals, it is often necessary to normalize signals of different scales to a unified range, which increases the hyperparameter search space and complicates the tuning process.

On the other hand, for different circuits, even the same metric could vary significantly in magnitude due to differences in the number of modules, nets, chip dimensions, and other factors. As a result, it is necessary to adjust the coefficients for the reward signals for each circuit individually, which further complicates cross-circuit fine-tuning.

To address this issue, we employ a self-adaptive shaping method to normalize reward signals into a reasonable range, thereby stabilizing the training process. Specifically, each reward component is normalized based on circuit-specific statistics or intrinsic geometric properties, ensuring that all signals contribute comparably to the overall reward function regardless of their original scales. This normalization mitigates the dominance of any single metric due to scale disparities, reduces the sensitivity to hyperparameter choices, and facilitates more efficient and robust policy optimization. Furthermore, by adaptively adjusting the normalization scheme for each circuit, our approach enhances the generalizability and transferability of the learned policy across diverse circuit instances.

- **Block-Terminal Distance:** This metric is normalized by the width and height of the chip canvas as follows:

$$d = \frac{d}{\frac{W+H}{2}}.$$

(15)

- **Block-Block Adjacency Length:** It is normalized by the average side length of the blocks:

$$l = \frac{l}{\sqrt{\frac{\sum_{i=1}^{n} a_i}{n}}},$$

(16)

where $a_i$ denotes the area of block $i$, and $n$ is the total number of blocks.

- **Alignment Score:** According to Definition B.1, this metric naturally falls within the range $[0, 1]$, and thus requires no further normalization.

- **Overlap:** The overlap is normalized by the average block area as follows:

$$o = \frac{o}{\frac{\sum_{i=1}^{n} a_i}{n}}.$$

(17)

- **HPWL:** Prior to training, we use current policy checkpoint to generate floorplan solutions and compute the average HPWL value, denoted as $\bar{w}$. The HPWL is then normalized as:

$$\text{HPWL} = \frac{\text{HPWL}}{\bar{w}}.$$

(18)

After normalization, all reward signals are brought into a reasonable and stable range (approximately confined to the $[0, 1]$ range), which facilitates the subsequent training process. Moreover, these signals are adaptively adjusted across different circuits, which is also advantageous for fine-tuning among various circuit designs.

**Reward Function.** We design our reward function for 3D floorplanning with complex hardware design rules based on the one used in FlexPlanner (Zhong et al., 2024). Sparse episodic rewards often cause the training process to stagnate, leading to suboptimal performance and inefficient sample complexity. To address this issue, a dense reward scheme is selected. At the end of each episode, we compute the baseline $b$. For intermediate steps, the reward is calculated as the difference in metrics between consecutive steps, adjusted by adding the baseline $b$. The details of the reward design are provided in Alg. 1.

---

**Algorithm 1** Reward function for 3D floorplanning with complex hardware design rules.

---

1: **Input:** Normalized alignment score aln, overlap $o$, HPWL, block-block adjacency length $l$, block-terminal distance $d$, and corresponding weights $w_a$, $w_o$, $w_{\text{HPWL}}$, $w_l$, $w_d$
2: **Output:** Reward $r$ for each step
3: **for** $t = \text{len}(episode)$ **to** 1 **do**
4:     **if** $t$ is the end of an episode **then**
5:         $r_t \leftarrow w_a \cdot \text{aln}_t - w_o \cdot o_t - w_{\text{HPWL}} \cdot \text{HPWL}_t + w_l \cdot l_t - w_d \cdot d_t$
6:         $b \leftarrow r_t$ **// calculate baseline $b$ at the end of each episode**
7:     **else**
8:         $r_t \leftarrow w_a \cdot (\text{aln}_t - \text{aln}_{t-1}) - w_o \cdot (o_t - o_{t-1}) - w_{\text{HPWL}} \cdot (\text{HPWL}_t - \text{HPWL}_{t-1})$
9:         $r_t \leftarrow r_t + w_l \cdot (l_t - l_{t-1}) - w_d \cdot (d_t - d_{t-1})$
10:        $r_t \leftarrow r_t + b$ **// add baseline $b$ for all intermediate steps**
11:     **end if**
12: **end for**

---

### C.2. Construction of Adjacent Terminal Mask

The procedure to construct the adjacent terminal mask mentioned in Sec. 4.1 is shown in Alg. 2. Given a movable block $b$ and its adjacent terminal $t$, we place $b$ on all possible positions, and calculate the distance between terminal $t$ and four edges of block $b$. The minimum value of these four distances is viewed as the block-terminal distance.

### C.3. Construction of Adjacent Block Mask

The procedure to construct the adjacent block mask mentioned in Sec. 4.1 is shown in Alg. 3. To fully exploit the parallel computing capabilities of GPUs and expedite the construction process, vectorized operations are utilized.

## D. Training Pipeline

The overall training pipeline is shown in Alg. 4, based on PPO (Schulman et al., 2017) algorithm with RL framework tianshou (Weng et al., 2022). During a training epoch, the policy network collects training data into replay buffer by interacting with the environment. To accelerate the training process, the policy network simultaneously interacts with $n_e$ environments in parallel. We set the buffer size to satisfy

$$L_{buf} \equiv 0 \mod \left( n_e \times \text{len}\left( episode \right) \right), \tag{19}$$

ensuring that each episode in the replay buffer is completed and terminated with its final step, where $L_{buf}$ is the size of replay buffer. After data collection, we further compute the reward, cumulative discounted reward and advantage. Since each episode is completed in the replay buffer, each step has the corresponding terminated step (end step) within the same episode to calculate the reward in Alg. 1. Finally, both critic network and policy network are updated.

**In the training-from-scratch scenario,** both the policy and critic networks are randomly initialized for a given circuit and trained using a policy gradient-based algorithm, such as PPO. Data collection is achieved through the interaction between the policy network and the environment, eliminating the need for a pre-constructed offline dataset with expert floorplan data. This approach alleviates the scarcity of floorplan data and reduces the cost of acquiring expert floorplan layouts from human annotators.

---

**Algorithm 2** Construction of Adjacent Terminal Mask.

---

1: **Input:** Block to place $b$, adjacent terminal $t$ of block $b$
2: **Output:** Adjacent terminal mask $\boldsymbol{T}^{(bt)}$ with shape $(W, H)$
3: Let $x_t, y_t$ be the coordinates of the terminal, $w, h$ be the width and height of the block
4: Let $x_b \leftarrow \text{arange}(W), y_b \leftarrow \text{arange}(H)$, be a range of possible x-, y-coordinates for block positions
5: Reassign $x_b, y_b$ be the *meshgrid* of $x_b$ and $y_b$: $x_b, y_b = \text{meshgrid}(x_b, y_b, \text{indexing} = \text{ij})$
6: Let $ds$ be an empty tensor with shape $(4, W, H)$ to store distances between each block edge and terminal
7: **// For the bottom edge**
8: $x \leftarrow x_b.unsqueeze(-1) + \text{arange}(w).reshape(1, 1, w)$
9: $y \leftarrow y_b.unsqueeze(-1)$
10: $ds[0] \leftarrow \min(|x_t - x| + |y_t - y|, \dim = -1)$
11: **// For the top edge**
12: $x \leftarrow x_b.unsqueeze(-1) + \text{arange}(w).reshape(1, 1, w)$
13: $y \leftarrow y_b.unsqueeze(-1) + h - 1$
14: $ds[1] \leftarrow \min(|x_t - x| + |y_t - y|, \dim = -1)$
15: **// For the left edge**
16: $x \leftarrow x_b.unsqueeze(-1)$
17: $y \leftarrow y_b.unsqueeze(-1) + \text{arange}(h).reshape(1, 1, h)$
18: $ds[2] \leftarrow \min(|x_t - x| + |y_t - y|, \dim = -1)$
19: **// For the right edge**
20: $x \leftarrow x_b.unsqueeze(-1) + w - 1$
21: $y \leftarrow y_b.unsqueeze(-1) + \text{arange}(h).reshape(1, 1, h)$
22: $ds[3] \leftarrow \min(|x_t - x| + |y_t - y|, \dim = -1)$
23: **// Combine the minimum distances**
24: $mask \leftarrow \min(ds, \dim = 0)$
25: **Return:** $mask$

---

**In the fine-tuning scenario,** a target circuit $c_t$ is considered as an example. Specifically, after training the policy and critic networks on the source circuit $c_s$, the trained network checkpoints are used to instantiate new policy and critic networks for the target circuit $c_t$. The weights from the source circuit networks are loaded into the new networks, which are subsequently trained using an online RL algorithm. The training procedure then follows the same steps as in the training-from-scratch scenario.

## E. More Discussion about Extensibility

In this section, we illustrate the extensibility of our approach to accommodate emerging design constraints in future chip design. We provide two representative examples: one with an explicit constraint and another with an implicit constraint.

**Example 1 (Explicit Constraint: Block Distance)** Consider a new design rule where, given a fixed/placed block $b_0$ with shape $(h_0, w_0)$ and coordinate $(x_0, y_0)$, a movable block $b$ with shape $(h, w)$ must be placed such that the distance between $b_0$ and $b$ does not exceed a specified threshold:

$$d(b, b_0) \triangleq \left| x + \frac{w}{2} - x_0 - \frac{w_0}{2} \right| + \left| y + \frac{h}{2} - y_0 - \frac{h_0}{2} \right| \leq \bar{d}. \tag{20}$$

Our framework can be extended to incorporate this rule as follows:

1. Construct a rule matrix $\boldsymbol{D} \in \mathbb{R}^{W \times H}$ where

$$\boldsymbol{D}_{xy} = d(b(x, y), b_0), \tag{21}$$

   with $b$ placed at $(x, y)$. This matrix serves as an input feature to the policy network.

2. Define a binary mask $\bar{\boldsymbol{D}} \in \mathbb{R}^{W \times H}$ to indicate valid positions:

$$\bar{\boldsymbol{D}}_{xy} = \begin{cases} 1, & \boldsymbol{D}_{xy} \leq \bar{d}, \\ 0, & \boldsymbol{D}_{xy} > \bar{d}. \end{cases} \tag{22}$$

---

**Algorithm 3** Construction of Adjacent Block Mask.

---

1: **Input:** Block to place $b_1$, adjacent block $b_2$ of block $b_1$
2: **Output:** Adjacent block mask $\boldsymbol{B}^{(b_1,b_2)}$, indicating adjacency between $b_1$ and $b_2$
3: **Assert:** $b_2$ has been placed
4: Let $w_1, h_1$ be the width and height of block $b_1$, $w_2, h_2$ be the width and height of block $b_2$
5: Let $x_2, y_2$ be the coordinates of block $b_2$ on the grid
6: Initialize $mask$ as a 2D matrix of zeros with shape $(W, H)$
7:   // **block** $b_1$ **is on the left of** $b_2$
8: $x_1 \leftarrow x_2 - w_1$
9: $y_1^{start} \leftarrow \max(y_2 - h_1 + 1, 0), y_1^{end} \leftarrow \min(y_2 + h_2, H)$
10: **if** $x_1 \geq 0$ **then**
11:     $left_1 \leftarrow \text{arange}(y_1^{start}, y_1^{end}), right_1 \leftarrow left_1 + h_1$
12:     $left_2 \leftarrow$ array with shape like $left_1$ filled with $y_2, right_2 \leftarrow left_2 + h_2$
13:     $l \leftarrow \max(left_1, left_2), r \leftarrow \min(right_1, right_2)$
14:     $o \leftarrow \max(r - l, 0)$ // **overlap between edges**
15:     $mask[x_1, y_1^{start} : y_1^{end}] \leftarrow o$
16: **end if**
17:   // **block** $b_1$ **is on the right of** $b_2$
18: $x_1 \leftarrow x_2 + w_2$
19: $y_1^{start} \leftarrow \max(y_2 - h_1 + 1, 0), y_1^{end} \leftarrow \min(y_2 + h_2, H)$
20: **if** $x_1 < W$ **then**
21:     $left_1 \leftarrow \text{arange}(y_1^{start}, y_1^{end}), right_1 \leftarrow left_1 + h_1$
22:     $left_2 \leftarrow$ array with shape like $left_1$ filled with $y_2, right_2 \leftarrow left_2 + h_2$
23:     $l \leftarrow \max(left_1, left_2), r \leftarrow \min(right_1, right_2)$
24:     $o \leftarrow \max(r - l, 0)$
25:     $mask[x_1, y_1^{start} : y_1^{end}] \leftarrow o$
26: **end if**
27:   // **block** $b_1$ **is below** $b_2$
28: $y_1 \leftarrow y_2 - h_1$
29: $x_1^{start} \leftarrow \max(x_2 - w_1 + 1, 0), x_1^{end} \leftarrow \min(x_2 + w_2, W)$
30: **if** $y_1 \geq 0$ **then**
31:     $left_1 \leftarrow \text{arange}(x_1^{start}, x_1^{end}), right_1 \leftarrow left_1 + w_1$
32:     $left_2 \leftarrow$ array with shape like $left_1$ filled with $x_2, right_2 \leftarrow left_2 + w_2$
33:     $l \leftarrow \max(left_1, left_2), r \leftarrow \min(right_1, right_2)$
34:     $o \leftarrow \max(r - l, 0)$
35:     $mask[x_1^{start} : x_1^{end}, y_1] \leftarrow o$
36: **end if**
37:   // **block** $b_1$ **is above** $b_2$
38: $y_1 \leftarrow y_2 + h_2$
39: $x_1^{start} \leftarrow \max(x_2 - w_1 + 1, 0), x_1^{end} \leftarrow \min(x_2 + w_2, W)$
40: **if** $y_1 < H$ **then**
41:     $left_1 \leftarrow \text{arange}(x_1^{start}, x_1^{end}), right_1 \leftarrow left_1 + w_1$
42:     $left_2 \leftarrow$ array with shape like $left_1$ filled with $x_2, right_2 \leftarrow left_2 + w_2$
43:     $l \leftarrow \max(left_1, left_2), r \leftarrow \min(right_1, right_2)$
44:     $o \leftarrow \max(r - l, 0)$
45:     $mask[x_1^{start} : x_1^{end}, y_1] \leftarrow o$
46: **end if**
47: **Return:** $mask$

---

This mask can be integrated into the availability mask $\boldsymbol{M}$.

3. Incorporate the distance metric into the reward function to guide RL policy optimization.

---

**Algorithm 4** Training Algorithm.

---

**for** epoch **from** 1 **to** n_epochs **do**
  // data collection
  **while** replay buffer is not full **do**
    // action probability calculation
    calculate action probability for position of current block $b_t$: $\pi_\theta\left(a_t^{(1)}|s_t\right)$

    sample $(x_t, y_t) \sim \pi_\theta\left(a_t^{(1)}|s_t\right)$

    calculate action probability for aspect ratio of next block $b_{t+1}$: $\pi_\theta\left(a_t^{(2)}|s_t\right)$

    sample $\mathrm{AR}_{t+1} \sim \pi_\theta\left(a_t^{(2)}|s_t\right)$
    // action execution and next state observation
    execute the action $a_t$:
        place block $b_t$ at position $(x_t, y_t)$
        access next block $b_{t+1}$
        set the aspect ratio of $b_{t+1}$ to $\mathrm{AR}_{t+1}$
    observe next state $s_{t+1}$ from environment
    get alignment score $\mathrm{aln}_t$, wirelength $\mathrm{HPWL}_t$, block-block adjacency length $l_t$, block-terminal distance $d_t$, overlap $o_t$ from environment
    add $\left(s, s', (x, y, \mathrm{AR}), (\mathrm{aln}, \mathrm{HPWL}, l, d, o), (\pi_{\theta_{old}}^{(1)}, \pi_{\theta_{old}}^{(2)})\right)$ into buffer
  **end while**
  // data processing (state value computation)
  **for** each sample **in** buffer **do**
    re-compute reward $r$ according to Alg. 1
    calculate state value for current state $s$: $v = V_\phi(s)$
    calculate state value for next state $s'$: $v' = V_\phi(s')$
    modify current sample with adding $(r, v, v')$
  **end for**
  // data processing (advantage computation)
  **for** each sample **in** buffer **do**
    compute cumulative discounted reward $G$ and advantage $\hat{A}$ for current sample according to generalized advantage estimation (GAE) (Schulman et al., 2015; 2017)
    $\hat{A}_t = \sum_{i=0}^{T-t-1}(\gamma\lambda)^i \delta_{t+i}$, where $\delta_t = r_t + \gamma v_{t+1} - v_t$
    $G_t = \hat{A}_t + v_t$
    modify current sample by adding $(\hat{A}, G)$
  **end for**
  // network update
  **for** update_epoch **from** 1 **to** n_update_epochs **do**
    **for** minibatch **in** buffer **do**
      // policy/actor network update
      compute action probability $\pi_\theta^{(1)}, \pi_\theta^{(2)}$
      compute action distribution entropy $h_\theta^{(1)}, h_\theta^{(2)}$
      update policy/actor network according to Eq. 10
      // critic network update
      compute state value $v = V_\phi(s)$
      update critic network according to Eq. 11
    **end for**
  **end for**
**end for**

---

**Example 2 (Implicit Constraint: Thermal Optimization)** For design rules without hard constraints on block coordinates, the binary mask for valid positions can be omitted. Consider thermal optimization, where each block is associated with a

power value $p$ and the objective is to minimize the peak temperature. The procedure is as follows:

1. Construct a rule matrix $\boldsymbol{P} \in \mathbb{R}^{W \times H}$, i.e., the power map (Ziabari et al., 2014), where $\boldsymbol{P}_{xy}$ denotes the power density at $(x, y)$. The heat distribution map $\boldsymbol{H} \in \mathbb{R}^{W \times H}$, with $\boldsymbol{H}_{xy}$ representing temperature at $(x, y)$, can be computed using professional tools such as ANSYS (Madenci & Guven, 2015), or approximated by numerical (Zhan & Sapatnekar, 2005a) or convolution (Hériz et al., 2007) methods. Both $\boldsymbol{P}$ and $\boldsymbol{H}$ can be used as input features for the policy network to capture thermal characteristics.

2. Since the thermal constraint is implicit, the binary mask construction can be omitted.

3. The temperature distribution can be evaluated in step 1. As a result, metrics such as peak or average temperature can be incorporated into the reward function to guide RL training.

## F. Model Architecture

### F.1. Actor Network (Policy)

The policy network $\pi_\theta$ contains two parts: the encoder $E_\theta$ and decoder/generator $D_\theta$. Chip canvas layout $\boldsymbol{C}$, alignment mask $\boldsymbol{A}$, adjacent terminal mask $\boldsymbol{T}$, adjacent block mask $\boldsymbol{B}$, wire mask $\boldsymbol{W}$, position mask $\boldsymbol{P}$ are concatenated along the channel dimension. These stacked matrices $\boldsymbol{S}$ are fed into a CNN $E_\theta$ with several blocks. Each block consists of a convolutional layer with kernel size 3 and stride 1, a batch normalization layer, a GELU (Hendrycks & Gimpel, 2016) activation function, and a max pooling layer with kernel size 2 and stride 2. The netlist graph is processed by a two-layer Transformer Encoder, where node features are reshaped into a sequence and the adjacency matrix is used as the attention mask. Sinusoidal positional encoding is applied to the placement order of each block to incorporate positional information.

**For the position probability for all movable blocks,** we use a decoder $D_\theta$. The decoder $D_\theta$ contains several blocks, where each block consists of an upsampling layer with a scale factor 2, a convolutional block with kernel size 3 and stride 1, a batch normalization layer, and a GELU activation layer. Note that we use upsampling layer instead of transposed convolutional layer to realize scale-up to the original size $(W, H)$, where $W, H$ are the width and height of the chip canvas.

Besides, these stacked matrices are also fed into another CNN, local encoder $C_\theta$ with three blocks, where each block contains a convolutional layer with kernel size 1 and stride 1, and a GELU activation layer. Finally, the output of $C_\theta$ and the decoder $D_\theta$ are merged with a convolutional layer (the fusion module) to get the position probability matrix $p_\theta \in \mathbb{R}^{W \times H}$.

**For the aspect ratio for all soft blocks**, a Multilayer Perceptron (MLP) takes $E_\theta(\boldsymbol{S})$ as input and outputs the mean value $\mu_\theta$ and logarithm of standard deviation $\log \sigma_\theta$. These statistics are utilized to construct a Gaussian distribution to determine the aspect ratio AR.

### F.2. Critic Network

The critic network $V_\phi$ shares the parameters with the encoder $E_\theta$. Then a global max pooling layer is utilized to reduce the spatial dimension of $E_\theta(\boldsymbol{S})$, and the state value $V \in \mathbb{R}$ is computed through an MLP.

## G. Baselines

The baselines referred in Sec. 5 are introduced as follows, which can be categorized into heuristics-based and reinforcement learning-based approaches.

**Analytical method** (Huang et al., 2023) formulates fixed-outline floorplanning as an analytical optimization problem, minimizing an objective function that combines wirelength and a potential energy term to manage module overlap. It uses an analytical solution to Poisson's equation to define this potential energy, which allows the widths of soft modules to be treated as optimizable variables directly within the energy function. The algorithm performs global floorplanning to determine module positions and shapes by solving this model. We extend it to 3D configuration by applying global floorplanning separately on each layer.

**B*-3D-SA** (Shanthi et al., 2021) is a heuristic-based approach for 3D floorplanning. It represents each die of the 3D layout using a B*-tree (Chang et al., 2000). Simulated Annealing (SA) (Bertsimas & Tsitsiklis, 1993) is employed to search for an optimal solution. At each iteration, the current B*-tree is randomly perturbed to a new state, with the transition accepted

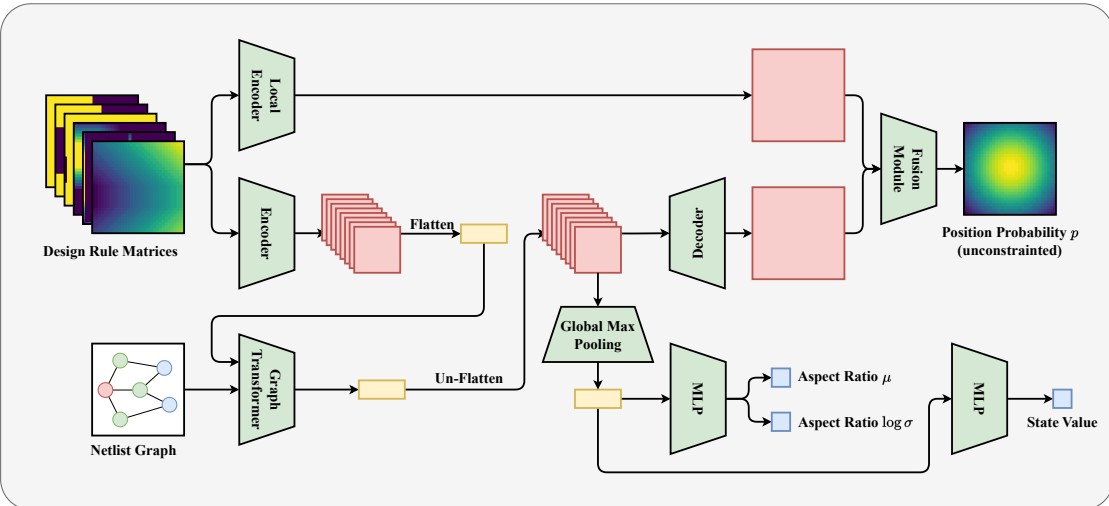

*Figure 6.* Model architecture of our approach. Actor (policy) and critic networks share the same Encoder network to process the matrices representations. Up-sampling blocks + convolutional blocks are used to realize up-scaling instead of transposed convolutional blocks.

based on a certain probability. The perturbation is achieved through one of the following operations: 1) modifying the aspect ratio of a block, 2) swapping two nodes within a tree, 3) relocating a node within the tree, or 4) relocating a sub-tree within the tree. Afterward, the B*-tree is decoded into the corresponding floorplan layout. In prior work, the cost function included only the Half-Perimeter Wire Length (HPWL) and an out-of-bounds penalty. To better account for the objectives in 3D floorplanning, we extend the cost function by incorporating additional factors: block-block adjacency length, block-terminal distance, and inter-die alignment score.

**GraphPlace** (Mirhoseini et al., 2021) is a reinforcement learning-based approach for 2D floorplanning task. It employs an edge-based graph convolutional neural network architecture to learn the representation of the netlist graph. It utilizes a sparse reward scheme, where the reward for intermediate steps is zero, and the final reward is based on the HPWL and overlap at the termination step. To incorporate hardware design rules for the 3D floorplanning task, additional factors such as the alignment score, block-block adjacency length, and block-terminal distance are included in the reward calculation, maintaining the same sparse scheme to align with the original reward function design.

**DeepPlace** (Cheng & Yan, 2021) is a reinforcement learning-based approach for 2D floorplanning task. One key design in its learning paradigm involves a multi-view embedding model to encode both global graph level and local node level information of the input macros, including netlist graph and floorplanning layout. Moreover, it adopts a dense reward scheme, where final metrics are involved in extrinsic reward calculation in terminate step, the random network distillation is utilized for intrinsic reward at intermediate steps. To incorporate hardware design rules for the 3D floorplanning task, additional factors such as the alignment score, block-block adjacency length, and block-terminal distance are included in the reward calculation, maintaining the same sparse scheme to align with the original reward function design.

**MaskPlace** (Lai et al., 2022) adopts reinforcement learning framework to tackle the 2D floorplanning task. It recasts the floorplanning task as a problem of learning pixel-level visual representation to comprehensively describe modules on a chip. It proposes the wire mask, a novel matrix representation for modeling HPWL. The policy network is guided through a dense reward function, which utilizes the difference of HPWL between adjacent steps. To incorporate hardware design rules for the 3D floorplanning task, additional factors such as the alignment score, block-block adjacency length, and block-terminal distance are included in the reward calculation, maintaining the same sparse scheme to align with the original reward function design.

**Wiremask-BBO** (Shi et al., 2023) is a black-box optimization (BBO) framework, by using a wire-mask-guided (Lai et al., 2022) greedy procedure for objective evaluation. At each step, it utilizes the position mask to filter out invalid positions causing overlap or out-of-bound. Within the valid positions, the location with the minimum increment of HPWL is selected to place current block. However, it neglects the adherence to other design constraints, such as inter-die block alignment, grouping and boundary constraints. As a result, optimization of corresponding metrics, including alignment

score, block-block adjacency length and block-terminal distance is not taken into consideration. Additionally, it is incapable of addressing the variable aspect ratio of soft blocks.

**FlexPlanner** (Zhong et al., 2024) is a reinforcement learning-based method to tackle the inter-die block alignment constraint in 3D floorplanning task. Alignment mask is proposed to model this rule. Besides, multi-modality representations are selected to depict the 3D floorplanning task, including floorplanning image vision, netlist graph, and placing order sequence. To incorporate hardware design rules for the 3D floorplanning task, additional factors such as the block-block adjacency length, and block-terminal distance are included in the reward calculation, maintaining the same sparse scheme to align with the original reward function design.

In conclusion, the aforementioned approaches are tailored to specific design rules. For instance, MaskPlace (Lai et al., 2022) is designed to prevent overlap between blocks, while FlexPlanner (Zhong et al., 2024) focuses on optimizing inter-die block alignment. Our method targets the simultaneous satisfaction of more than seven industrial design rules, addressing the complexities of real-world 3D IC design.

## H. Computational Resources

We use deep learning framework PyTorch (Paszke et al., 2019) and Reinforcement Learning framework tianshou (Weng et al., 2022). We select Adam (Kingma & Ba, 2014) optimizer with learning rate $1 \times 10^{-4}$. Training and testing are implemented on a Linux server with one NVIDIA GeForce RTX 4090 GPU with 24 GB CUDA memory, one AMD EPYC 7402 24-Core Processor at 3.35 GHz and 512 GB RAM.

## I. Settings of Hyper-Parameters

We list the settings of all hyper-parameters in Table 8.

## J. Constraint for Each Circuit

We list the statistics for each circuit in the evaluation benchmark MCNC and GSRC in Table 9. 'Adj.' means adjacent. '# Aln. Block' means the number of blocks which have alignment partners, '# Adj. Terminal' means the number of blocks with adjacent terminals to align with, '# Adj. Block' refers to the number of blocks that should be physically adjacent to other blocks.

We present these constraints along with their corresponding representations and penalties in Table 10. The representations, expressed as design rule matrices, encode essential information and features related to these constraints. These matrices are further employed to construct the availability mask, which filters out invalid positions that violate the design rules. The rewards and penalties associated with the constraints serve as guiding signals for optimizing the network.

## K. Runtime Evaluation

### K.1. Environment Transition

We evaluate the time required to construct each representation in the environment state transition process. As shown in Fig. 7a, the time spent constructing the adjacent block mask and terminal mask constitutes only a small portion of the overall state update process. By utilizing parallel computational operators, the processing power of GPU can be fully exploited to accelerate the construction of these matrices.

### K.2. Inference Time

We also compare the inference time for each method, which is shown in Fig. 7b. Inference time of our RulePlanner is either lower or comparable to other learn-based methods (Mirhoseini et al., 2021; Cheng & Yan, 2021; Lai et al., 2022; Zhong et al., 2024), and is better than the heuristic-based method (Shanthi et al., 2021), where tens of thousands of iterations are required. Pipeline of Wiremask-BBO (Shi et al., 2023) is the same as MaskPlace (Lai et al., 2022), while neural network is not involved in it, contributing to the shortest inference time.

*Table 8.* Hyper-parameters.

| Argument | Value |
|---|---|
| area utilization | 80 % |
| learning rate | 0.0001 |
| batch size | 128 |
| buffer size | $n_e \times \text{len}(episode)$ |
| parallel environments $n_e$ | 8 |
| number of epochs | 500/1,000 |
| number of update epochs | 10 |
| GAE $\lambda$ | 0.95 |
| clip $\varepsilon$ | 0.2 |
| clip loss weight for position decision $\lambda_1$ | 1.0 |
| clip loss weight for aspect ratio decision $\lambda_2$ | 0.5 |
| value loss weight $\lambda_\phi$ | 0.5 |
| die height $H$ | 128 |
| die width $W$ | 128 |
| number of dies $|\mathcal{D}|$ | 2 |
| range of block aspect ratio | $\left[\frac{1}{2}, 2\right]$ |
| reward discount factor $\gamma$ | 0.99 |
| reward weight for HPWL $w_{\text{HPWL}}$ | 1.0 |
| reward weight for alignment $w_a$ | 0.5 |
| reward weight for overlap $w_o$ | 0.5 |
| reward weight for block-block adjacent length $w_l$ | 4.0 |
| reward weight for block-terminal distance $w_d$ | 4.0 |
| threshold for adjacent terminal mask $\bar{t}$ | 0 |
| threshold for adjacent block mask $\bar{b}$ | 0 |
| threshold for alignment mask $\bar{a}$ between block $b_i, b_j$ | $0.1 \times \min\{a_i, a_j\}$ |

*Table 9.* Constraints for each circuit.

| Circuit | # Block | # Terminal | # Net | # Aln. Block | # Adj. Terminal | # Adj. Block |
|---|---|---|---|---|---|---|
| ami33 | 33 | 40 | 121 | 20 | 5 | 10 |
| ami49 | 49 | 22 | 396 | 20 | 5 | 10 |
| n10 | 10 | 69 | 118 | 10 | 5 | 10 |
| n30 | 30 | 212 | 349 | 20 | 5 | 10 |
| n50 | 50 | 209 | 485 | 20 | 5 | 10 |
| n100 | 100 | 334 | 885 | 60 | 10 | 20 |
| n200 | 200 | 564 | 1,585 | 60 | 10 | 20 |
| n300 | 300 | 569 | 1,893 | 60 | 10 | 20 |

*Table 10.* Design rules, corresponding matrices representation and rewards/penalties for optimization.

| Design Rule | Representation | Reward/Penalty |
|---|---|---|
| Boundary Constraint | Adjacent Terminal Mask | Block-terminal distance |
| Grouping Constraint | Adjacent Block Mask | Block-Block Adjacency Length |
| Inter-Die Block Alignment | Alignment Mask | Alignment Score |
| Non-Overlap & Fixed-Outline & Pre-Placement | Position Mask | Overlap |

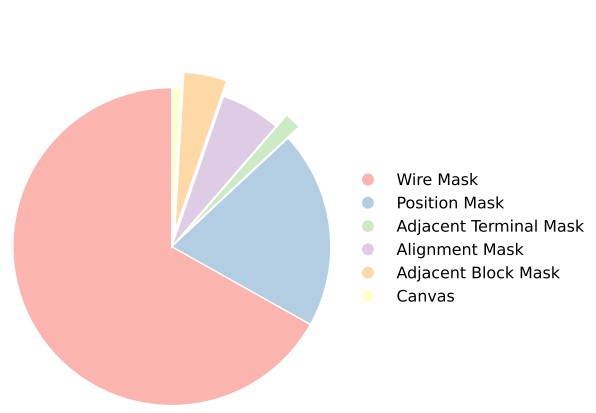

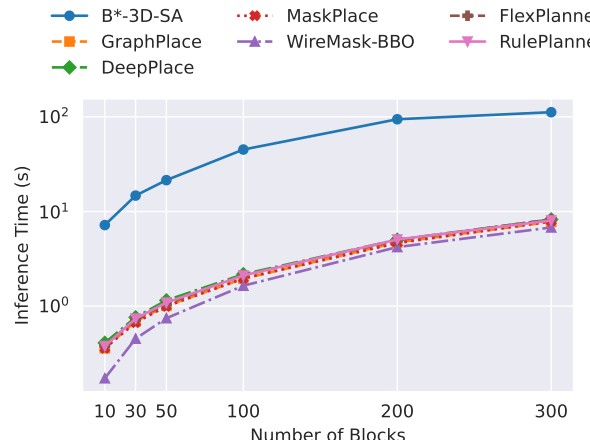

*(a)* Runtime breakdown for environment step. Leveraging the parallel construction of the adjacent block mask and terminal mask, they contribute only a small portion to the transition.

*(b)* Inference time comparison. Inference time of our RulePlanner is either lower or comparable to other learn-based methods, and is better than the heuristic-based method.

*Figure 7.* Runtime for environment transition and inference.

## L. More Results about the Main Experiments

We conduct additional main experiments (mentioned in Sec. 5.2 in the main paper) due to the page limitation. For each task with corresponding design rules listed in Table 2, we evaluate comprehensive metrics.

- Task 1 mainly focuses on (a) boundary constraint and (c) inter-die block alignment constraint. We evaluate the block-terminal distance, alignment score, wirelength and overlap in Table 3, 11, 12, and 13.

- Task 2 mainly focuses on (b) grouping constraint and (c) inter-die block alignment constraint. We evaluate the block-block adjacency length, alignment score, wirelength and overlap in Table 4, 14, 15 and 16.

- Task 3 mainly focuses on (a) boundary constraint, (b) grouping constraint, and (c) inter-die block alignment constraint. We evaluate the block-terminal distance, block-block adjacency length, alignment score, wirelength and overlap in Table 17, 18, 19, 20 and 21.

The metrics closely associated with design rules, such as block-terminal distance, block-block adjacency length and inter-die block alignment score, demonstrate that our approach outperforms all compared baseline methods. The framework for our approach discussed in Sec. 4.6 contributes to the effectiveness. Key components include matrix representations to model these design rules, constraints on the action space, and reward functions that guide optimization.

On some circuits, Wiremask-BBO results in slightly lower HPWL than our approach. This can be attributed to the fact that, during the placement decision process for each block, Wiremask-BBO uses a greedy strategy that selects positions minimizing HPWL increment without causing overlap. However, this approach ***overlooks other crucial hardware design rules***, such as boundary constraints, grouping constraints, and inter-die block alignment constraints. As a result, Wiremask-BBO fails to satisfy these higher-priority design rules, which are more significant than merely optimizing HPWL. In contrast, when comparing relevant metrics such as alignment score, block-block adjacency length, and block-terminal distance, our approach consistently outperforms Wiremask-BBO. This demonstrates that our method effectively incorporates these complex design rules during the floorplanning process.

We further demonstrate the 3D floorplan layout for Task 3, which incorporates complex hardware design rules, as illustrated in Fig. 8. The floorplan of the circuit *n100* with three dies/layers is presented.

- In Fig. 8a, the alignment constraint is demonstrated. Two blocks on different layers with the same index form an alignment pair. For a pair with block $i, j$, we calculate individual alignment score $\mathrm{aln}(i, j)$. Green means these two

*Table 11.* **Alignment Score** comparison on Task 1 among baselines and our method. The higher the Alignment Score, the better, and the optimal results are shown in **bold**. C/M means Circuit/Method.

| C/M | Analytical | B*-3D-SA | GraphPlace | DeepPlace | MaskPlace | WireMask-BBO | FlexPlanner | RulePlanner |
|---|---|---|---|---|---|---|---|---|
| ami33 | 0.120±0.039 | 0.809±0.170 | 0.079±0.036 | 0.115±0.001 | 0.196±0.018 | 0.116±0.019 | 0.656±0.026 | **0.947±0.004** |
| ami49 | 0.057±0.002 | 0.511±0.081 | 0.042±0.002 | 0.128±0.064 | 0.089±0.082 | 0.051±0.005 | 0.781±0.081 | **0.921±0.020** |
| n10 | 0.121±0.201 | 0.687±0.248 | 0.498±0.007 | 0.258±0.030 | 0.566±0.018 | 0.127±0.002 | 0.321±0.102 | **0.962±0.002** |
| n30 | 0.086±0.035 | 0.792±0.077 | 0.006±0.003 | 0.146±0.080 | 0.473±0.008 | 0.239±0.031 | 0.754±0.074 | **0.921±0.030** |
| n50 | 0.059±0.079 | 0.646±0.098 | 0.173±0.133 | 0.067±0.035 | 0.539±0.008 | 0.035±0.033 | 0.846±0.079 | **0.910±0.020** |
| n100 | 0.032±0.024 | 0.427±0.074 | 0.047±0.020 | 0.032±0.009 | 0.074±0.030 | 0.051±0.000 | 0.678±0.046 | **0.903±0.027** |
| n200 | 0.010±0.015 | 0.134±0.034 | 0.031±0.019 | 0.000±0.000 | 0.037±0.006 | 0.023±0.016 | 0.797±0.019 | **0.919±0.018** |
| n300 | 0.004±0.007 | 0.004±0.007 | 0.006±0.003 | 0.033±0.027 | 0.052±0.014 | 0.041±0.012 | 0.731±0.004 | **0.911±0.009** |
| Avg. | 0.061 | 0.501 | 0.110 | 0.097 | 0.253 | 0.085 | 0.696 | **0.924** |

*Table 12.* **HPWL** comparison on Task 1 among baselines and our method. The lower the HPWL, the better, and the optimal results are shown in **bold**. C/M means Circuit/Method. While Analytical and WireMask-BBO achieve lower HPWL on some circuits (n30 - n300), they demonstrate significantly poorer performance on other metrics like alignment rate, block-block adjacency length, and block-terminal distance, owing to their inability to fully satisfy complex design rules.

| C/M | Analytical | B*-3D-SA | GraphPlace | DeepPlace | MaskPlace | WireMask-BBO | FlexPlanner | RulePlanner |
|---|---|---|---|---|---|---|---|---|
| ami33 | 64,990±9,614 | 69,850±2,827 | 72,553±2,766 | 62,660±2,935 | 72,274±7,661 | 60,883±1,836 | 69,714±3,121 | **58,452±381** |
| ami49 | 1,288,030±29,714 | 955,853±29,229 | 1,087,753±60,066 | 1,155,578±47,067 | 1,248,628±9,358 | 1,042,248±95,424 | 889,463±47,435 | **796,524±900** |
| n10 | 26,829±370 | 31,718±1,050 | 32,776±267 | 27,507±1,075 | 31,344±117 | **26,207±28** | 29,939±85 | 31,498±68 |
| n30 | **79,912±50** | 93,123±3,515 | 86,325±650 | 87,751±220 | 101,341±293 | 85,556±773 | 93,498±1,544 | 94,755±1,196 |
| n50 | **106,127±946** | 122,526±2,887 | 116,490±1,878 | 117,302±133 | 121,726±558 | 109,575±1,615 | 118,458±1,270 | 118,593±551 |
| n100 | **161,774±598** | 210,401±7,370 | 197,022±3,162 | 185,625±392 | 193,759±485 | 177,365±1,389 | 191,060±1,093 | 197,014±1,804 |
| n200 | **290,904±1,133** | 384,706±7,498 | 400,404±6,406 | 371,355±762 | 377,917±1,049 | 313,249±1,230 | 343,771±287 | 331,904±1,892 |
| n300 | **407,349±2,521** | 635,742±4,148 | 583,817±1,149 | 550,687±10,817 | 651,191±9,932 | 447,035±1,070 | 499,726±3,570 | 458,125±3,650 |
| Avg. | 303,239 | 312,990 | 322,143 | 319,808 | 349,773 | 282,765 | 279,454 | **260,858** |

*Table 13.* **Overlap** comparison on Task 1 among baselines and our method. The lower the Overlap, the better, and the optimal results are shown in **bold**. C/M means Circuit/Method.

| C/M | Analytical | B*-3D-SA | GraphPlace | DeepPlace | MaskPlace | WireMask-BBO | FlexPlanner | RulePlanner |
|---|---|---|---|---|---|---|---|---|
| ami33 | 0.013±0.007 | **0.000±0.000** | 0.122±0.002 | 0.073±0.019 | 0.022±0.014 | 0.063±0.017 | 0.019±0.004 | **0.000±0.000** |
| ami49 | 0.011±0.002 | **0.000±0.000** | 0.000±0.000 | 0.008±0.004 | 0.002±0.002 | **0.000±0.000** | 0.001±0.001 | **0.000±0.000** |
| n10 | 0.027±0.021 | **0.000±0.000** | 0.094±0.005 | 0.242±0.026 | 0.122±0.001 | 0.189±0.000 | 0.087±0.057 | **0.000±0.000** |
| n30 | 0.025±0.009 | **0.000±0.000** | 0.031±0.001 | 0.078±0.011 | **0.000±0.000** | 0.004±0.006 | 0.036±0.028 | **0.000±0.000** |
| n50 | 0.007±0.003 | **0.000±0.000** | 0.010±0.005 | 0.019±0.001 | **0.000±0.000** | **0.000±0.000** | 0.002±0.002 | **0.000±0.000** |
| n100 | 0.014±0.006 | **0.000±0.000** | **0.000±0.000** | 0.003±0.000 | 0.001±0.001 | **0.000±0.000** | **0.000±0.000** | **0.000±0.000** |
| n200 | 0.050±0.002 | **0.000±0.000** | 0.001±0.001 | 0.000±0.000 | **0.000±0.000** | **0.000±0.000** | **0.000±0.000** | **0.000±0.000** |
| n300 | 0.042±0.006 | **0.000±0.000** | **0.000±0.000** | 0.000±0.000 | **0.000±0.000** | **0.000±0.000** | 0.001±0.001 | **0.000±0.000** |
| Avg. | 0.024 | **0.000** | 0.032 | 0.053 | 0.018 | 0.032 | 0.018 | **0.000** |

*Table 14.* **Alignment Score** comparison on Task 2 among baselines and our method. The higher the Alignment Score, the better, and the optimal results are shown in **bold**. C/M means Circuit/Method.

| C/M | Analytical | B*-3D-SA | GraphPlace | DeepPlace | MaskPlace | WireMask-BBO | FlexPlanner | RulePlanner |
|---|---|---|---|---|---|---|---|---|
| ami33 | 0.120±0.039 | 0.743±0.212 | 0.067±0.020 | 0.089±0.003 | 0.061±0.004 | 0.144±0.039 | 0.850±0.041 | **0.909±0.016** |
| ami49 | 0.057±0.002 | 0.452±0.021 | 0.090±0.027 | 0.065±0.039 | 0.047±0.008 | 0.094±0.056 | 0.759±0.055 | **0.910±0.016** |
| n10 | 0.121±0.201 | 0.733±0.167 | 0.164±0.043 | 0.181±0.023 | 0.432±0.004 | 0.186±0.001 | 0.802±0.132 | **0.954±0.003** |
| n30 | 0.086±0.035 | 0.788±0.096 | 0.122±0.018 | 0.087±0.075 | 0.391±0.008 | 0.241±0.036 | 0.704±0.069 | **0.936±0.018** |
| n50 | 0.059±0.079 | 0.655±0.088 | 0.063±0.026 | 0.029±0.013 | 0.280±0.041 | 0.008±0.011 | 0.808±0.073 | **0.968±0.001** |
| n100 | 0.032±0.024 | 0.363±0.074 | 0.032±0.013 | 0.031±0.028 | 0.159±0.006 | 0.037±0.008 | 0.859±0.027 | **0.959±0.005** |
| n200 | 0.010±0.015 | 0.134±0.034 | 0.019±0.017 | 0.009±0.002 | 0.251±0.009 | 0.019±0.014 | 0.757±0.029 | **0.939±0.016** |
| n300 | 0.004±0.007 | 0.004±0.007 | 0.025±0.025 | 0.003±0.003 | 0.180±0.005 | 0.012±0.006 | 0.822±0.027 | **0.915±0.014** |
| Avg. | 0.061 | 0.484 | 0.073 | 0.062 | 0.225 | 0.092 | 0.795 | **0.936** |

blocks are aligned ($\mathrm{aln}(i, j) \geq 0.5$) while red means not aligned ($\mathrm{aln}(i, j) < 0.5$). In this case, all pairs are aligned, meaning that blocks within a pair roughly share the same location across different dies.

- In Fig. 8b, the grouping constraint is shown. Two blocks on the same die with the same index form a group, which should be physically abutted. In this case, all groups satisfy this constraint, meaning that blocks within the same group share a common edge.

*Table 15.* **HPWL** comparison on Task 2 among baselines and our method. The lower the HPWL, the better, and the optimal results are shown in **bold**. C/M means Circuit/Method. While Analytical and WireMask-BBO achieve lower HPWL on some circuits (n30 - n300), they demonstrate significantly poorer performance on other metrics like alignment rate, block-block adjacency length, and block-terminal distance, owing to their inability to fully satisfy complex design rules.

| C/M | Analytical | B*-3D-SA | GraphPlace | DeepPlace | MaskPlace | WireMask-BBO | FlexPlanner | RulePlanner |
|---|---|---|---|---|---|---|---|---|
| ami33 | 64,990±9,614 | 65,502±4,778 | 65,778±3,629 | 72,868±2,650 | **61,185±677** | 67,030±4,837 | 62,942±1,424 | 64,941±1,623 |
| ami49 | 1,288,030±29,714 | 971,078±18,336 | 1,135,629±23,846 | 1,170,378±16,234 | 1,251,809±24,122 | 1,077,990±17,576 | 962,326±55,543 | **802,039±5,791** |
| n10 | 26,829±370 | 31,919±1,401 | **25,827±18** | 25,856±295 | 31,313±66 | 26,250±12 | 32,059±199 | 32,572±223 |
| n30 | **79,912±50** | 90,374±2,030 | 87,533±507 | 90,614±411 | 94,471±1,614 | 86,676±494 | 98,021±225 | 89,373±282 |
| n50 | 106,127±946 | 123,638±1,859 | 111,104±1,212 | 115,931±1,468 | 135,131±477 | 110,928±4,863 | 114,273±2,223 | 113,974±491 |
| n100 | **161,774±598** | 207,596±3,781 | 186,521±671 | 185,158±2,450 | 194,369±956 | 178,199±1,617 | 180,333±3,369 | 192,799±888 |
| n200 | **290,904±1,133** | 384,706±7,498 | 360,885±2,357 | 356,804±4,877 | 381,425±50 | 311,526±620 | 341,738±5,347 | 334,951±1,053 |
| n300 | **407,349±2,521** | 635,742±4,148 | 520,280±1,134 | 515,107±1,465 | 554,153±708 | 449,249±979 | 484,862±5,205 | 493,287±2,893 |
| Avg. | 303,239 | 313,819 | 311,695 | 316,590 | 337,982 | 288,481 | 284,569 | **265,492** |

*Table 16.* **Overlap** comparison on Task 2 among baselines and our method. The lower the Overlap, the better, and the optimal results are shown in **bold**. C/M means Circuit/Method.

| C/M | Analytical | B*-3D-SA | GraphPlace | DeepPlace | MaskPlace | WireMask-BBO | FlexPlanner | RulePlanner |
|---|---|---|---|---|---|---|---|---|
| ami33 | 0.013±0.007 | **0.000±0.000** | 0.021±0.004 | 0.035±0.007 | 0.010±0.006 | 0.035±0.034 | **0.000±0.000** | **0.000±0.000** |
| ami49 | 0.011±0.002 | **0.000±0.000** | 0.002±0.001 | 0.003±0.003 | 0.006±0.000 | **0.000±0.000** | **0.000±0.000** | **0.000±0.000** |
| n10 | 0.027±0.021 | **0.000±0.000** | 0.204±0.021 | 0.183±0.003 | 0.176±0.000 | 0.189±0.000 | 0.043±0.030 | **0.000±0.000** |
| n30 | 0.025±0.009 | **0.000±0.000** | 0.034±0.011 | 0.026±0.007 | 0.047±0.007 | 0.018±0.017 | 0.029±0.028 | **0.000±0.000** |
| n50 | 0.007±0.003 | **0.000±0.000** | 0.001±0.001 | 0.027±0.021 | 0.004±0.001 | **0.000±0.000** | 0.002±0.002 | **0.000±0.000** |
| n100 | 0.014±0.006 | **0.000±0.000** | 0.002±0.002 | 0.001±0.001 | **0.000±0.000** | **0.000±0.000** | **0.000±0.000** | **0.000±0.000** |
| n200 | 0.050±0.002 | **0.000±0.000** | **0.000±0.000** | **0.000±0.000** | **0.000±0.000** | **0.000±0.000** | **0.000±0.000** | **0.000±0.000** |
| n300 | 0.042±0.006 | **0.000±0.000** | **0.000±0.000** | **0.000±0.000** | **0.000±0.000** | **0.000±0.000** | 0.001±0.001 | **0.000±0.000** |
| Avg. | 0.024 | **0.000** | 0.033 | 0.034 | 0.030 | 0.030 | 0.009 | **0.000** |

*Table 17.* **Block-Terminal Distance** comparison on Task 3 among baselines and our method. The lower the Block-Terminal Distance, the better, and the optimal results are shown in **bold**. C/M means Circuit/Method.

| C/M | Analytical | B*-3D-SA | GraphPlace | DeepPlace | MaskPlace | WireMask-BBO | FlexPlanner | RulePlanner |
|---|---|---|---|---|---|---|---|---|
| ami33 | 0.079±0.058 | 0.164±0.156 | 0.382±0.174 | 0.475±0.080 | 0.109±0.021 | 0.514±0.002 | 0.220±0.004 | **0.000±0.000** |
| ami49 | 0.101±0.067 | 0.192±0.098 | 0.443±0.062 | 0.429±0.049 | 0.461±0.031 | 0.580±0.124 | 0.502±0.002 | **0.000±0.000** |
| n10 | 0.237±0.172 | 0.186±0.176 | 0.004±0.004 | 0.068±0.014 | 0.004±0.000 | 0.857±0.003 | 0.197±0.018 | **0.000±0.000** |
| n30 | 0.195±0.092 | 0.186±0.062 | 0.562±0.029 | 0.492±0.012 | 0.299±0.001 | 0.730±0.002 | 0.198±0.033 | **0.000±0.000** |
| n50 | 0.230±0.008 | 0.187±0.166 | 0.633±0.056 | 0.325±0.023 | 0.470±0.134 | 0.598±0.052 | 0.380±0.009 | **0.000±0.000** |
| n100 | 0.169±0.016 | 0.310±0.056 | 0.791±0.105 | 0.740±0.110 | 0.331±0.110 | 0.730±0.043 | 0.268±0.005 | **0.000±0.000** |
| n200 | 0.188±0.012 | 0.606±0.141 | 0.739±0.172 | 0.798±0.039 | 0.408±0.021 | 0.694±0.003 | 0.511±0.043 | **0.000±0.000** |
| n300 | 0.150±0.023 | 0.789±0.090 | 0.778±0.055 | 0.801±0.036 | 0.553±0.103 | 0.675±0.011 | 0.344±0.007 | **0.000±0.000** |
| Avg. | 0.169 | 0.328 | 0.542 | 0.516 | 0.329 | 0.672 | 0.327 | **0.000** |

*Table 18.* **Block-Block Adjacency Length** comparison on Task 3 among baselines and our method. The higher the Block-Block Adjacency Length, the better, and the optimal results are shown in **bold**. C/M means Circuit/Method.

| C/M | Analytical | B*-3D-SA | GraphPlace | DeepPlace | MaskPlace | WireMask-BBO | FlexPlanner | RulePlanner |
|---|---|---|---|---|---|---|---|---|
| ami33 | 0.000±0.000 | 0.030±0.026 | 0.000±0.000 | 0.000±0.000 | 0.057±0.012 | 0.083±0.018 | 0.051±0.021 | **0.210±0.004** |
| ami49 | 0.000±0.000 | 0.006±0.011 | 0.041±0.009 | 0.016±0.016 | 0.034±0.016 | 0.024±0.034 | 0.021±0.009 | **0.172±0.003** |
| n10 | 0.000±0.000 | 0.207±0.180 | **0.219±0.000** | 0.000±0.000 | 0.195±0.008 | **0.219±0.000** | 0.000±0.000 | 0.137±0.012 |
| n30 | 0.000±0.000 | 0.058±0.043 | 0.039±0.003 | 0.034±0.009 | 0.053±0.053 | 0.066±0.030 | 0.011±0.011 | **0.191±0.005** |
| n50 | 0.020±0.017 | 0.021±0.036 | 0.000±0.000 | 0.014±0.014 | 0.036±0.036 | 0.014±0.020 | 0.000±0.000 | **0.216±0.001** |
| n100 | 0.004±0.007 | 0.007±0.009 | 0.011±0.011 | 0.005±0.003 | 0.001±0.001 | 0.026±0.004 | 0.008±0.008 | **0.147±0.000** |
| n200 | 0.000±0.000 | 0.003±0.006 | 0.005±0.005 | 0.005±0.005 | 0.000±0.000 | 0.004±0.005 | 0.000±0.000 | **0.119±0.003** |
| n300 | 0.000±0.000 | 0.000±0.000 | 0.009±0.009 | 0.006±0.006 | 0.000±0.000 | 0.000±0.000 | 0.000±0.000 | **0.101±0.004** |
| Avg. | 0.003 | 0.042 | 0.040 | 0.010 | 0.047 | 0.054 | 0.011 | **0.162** |

- In Fig. 8c, the boundary constraint is illustrated. A block should align with a specific terminal on the boundary. Green means they are aligned while red means not aligned. In this example, all blocks and terminals are properly aligned.

*Table 19.* **Alignment Score** comparison on Task 3 among baselines and our method. The higher the Alignment Score, the better, and the optimal results are shown in **bold**. C/M means Circuit/Method.

| C/M | Analytical | B*-3D-SA | GraphPlace | DeepPlace | MaskPlace | WireMask-BBO | FlexPlanner | RulePlanner |
|---|---|---|---|---|---|---|---|---|
| ami33 | 0.120±0.039 | 0.743±0.212 | 0.084±0.026 | 0.105±0.046 | 0.113±0.050 | 0.028±0.008 | 0.567±0.014 | **0.937±0.016** |
| ami49 | 0.057±0.002 | 0.452±0.021 | 0.068±0.061 | 0.005±0.000 | 0.023±0.022 | 0.027±0.003 | 0.836±0.004 | **0.937±0.005** |
| n10 | 0.121±0.201 | 0.733±0.167 | 0.432±0.004 | 0.279±0.003 | 0.533±0.001 | 0.186±0.001 | 0.850±0.090 | **0.940±0.013** |
| n30 | 0.086±0.035 | 0.788±0.096 | 0.166±0.068 | 0.095±0.045 | 0.466±0.004 | 0.066±0.008 | 0.811±0.091 | **0.914±0.031** |
| n50 | 0.059±0.079 | 0.655±0.088 | 0.078±0.057 | 0.153±0.063 | 0.225±0.092 | 0.129±0.045 | 0.745±0.093 | **0.983±0.002** |
| n100 | 0.032±0.024 | 0.363±0.074 | 0.052±0.024 | 0.014±0.013 | 0.055±0.006 | 0.040±0.015 | 0.725±0.067 | **0.924±0.018** |
| n200 | 0.010±0.015 | 0.134±0.034 | 0.018±0.016 | 0.000±0.000 | 0.054±0.002 | 0.029±0.002 | 0.731±0.013 | **0.956±0.007** |
| n300 | 0.004±0.007 | 0.004±0.007 | 0.000±0.000 | 0.001±0.000 | 0.075±0.015 | 0.017±0.021 | 0.726±0.000 | **0.913±0.008** |
| Avg. | 0.061 | 0.484 | 0.112 | 0.081 | 0.193 | 0.065 | 0.749 | **0.938** |

*Table 20.* **HPWL** comparison on Task 3 among baselines and our method. The lower the HPWL, the better, and the optimal results are shown in **bold**. C/M means Circuit/Method. While Analytical and WireMask-BBO achieve lower HPWL on some circuits (n30 - n300), they demonstrate significantly poorer performance on other metrics like alignment rate, block-block adjacency length, and block-terminal distance, owing to their inability to fully satisfy complex design rules.

| C/M | Analytical | B*-3D-SA | GraphPlace | DeepPlace | MaskPlace | WireMask-BBO | FlexPlanner | RulePlanner |
|---|---|---|---|---|---|---|---|---|
| ami33 | 64,990±9,614 | 65,502±4,778 | **60,255±856** | 68,402±4,413 | 62,919±611 | 65,522±3,672 | 61,381±2,467 | 62,892±571 |
| ami49 | 1,288,030±29,714 | 971,078±18,336 | 1,177,953±4,283 | 1,107,628±98,289 | 1,078,211±13,164 | 1,191,798±57,462 | 914,486±39,216 | **887,036±9,156** |
| n10 | 26,829±370 | 31,919±1,401 | 30,993±334 | 33,374±327 | 30,892±23 | **26,250±12** | 29,350±206 | 32,136±162 |
| n30 | **79,912±50** | 90,374±2,030 | 88,632±224 | 91,353±750 | 98,230±456 | 89,361±909 | 94,824±1,135 | 94,997±671 |
| n50 | **106,127±946** | 123,638±1,859 | 115,108±3,615 | 124,905±71 | 121,115±1,761 | 109,583±1,918 | 119,264±1,325 | 120,204±352 |
| n100 | **161,774±598** | 207,596±3,781 | 195,964±3,061 | 185,501±909 | 203,878±5,279 | 178,027±3,913 | 197,508±4,519 | 197,302±346 |
| n200 | **290,904±1,133** | 384,706±7,498 | 396,816±2,440 | 364,732±731 | 383,161±6,764 | 312,416±1,433 | 356,295±22 | 345,462±204 |
| n300 | **407,349±2,521** | 635,742±4,148 | 626,088±2,487 | 624,785±2,866 | 606,651±9,411 | 456,588±4,291 | 510,457±2,048 | 462,643±2,884 |
| Avg. | 303,239 | 313,819 | 336,476 | 325,085 | 323,132 | 303,693 | 285,446 | **275,334** |

*Table 21.* **Overlap** comparison on Task 3 among baselines and our method. The lower the Overlap, the better, and the optimal results are shown in **bold**. C/M means Circuit/Method.

| C/M | Analytical | B*-3D-SA | GraphPlace | DeepPlace | MaskPlace | WireMask-BBO | FlexPlanner | RulePlanner |
|---|---|---|---|---|---|---|---|---|
| ami33 | 0.013±0.007 | **0.000±0.000** | 0.087±0.017 | 0.033±0.000 | 0.017±0.011 | 0.041±0.038 | 0.005±0.005 | **0.000±0.000** |
| ami49 | 0.011±0.002 | **0.000±0.000** | 0.007±0.006 | 0.002±0.000 | 0.001±0.001 | 0.003±0.003 | 0.000±0.000 | **0.000±0.000** |
| n10 | 0.027±0.021 | **0.000±0.000** | 0.114±0.016 | 0.059±0.037 | 0.095±0.000 | 0.189±0.000 | 0.116±0.040 | **0.000±0.000** |
| n30 | 0.025±0.009 | **0.000±0.000** | 0.032±0.030 | 0.047±0.011 | 0.003±0.000 | 0.003±0.004 | 0.009±0.000 | **0.000±0.000** |
| n50 | 0.007±0.003 | **0.000±0.000** | 0.005±0.002 | 0.011±0.008 | 0.006±0.006 | **0.000±0.000** | 0.004±0.000 | **0.000±0.000** |
| n100 | 0.014±0.006 | **0.000±0.000** | 0.002±0.000 | **0.000±0.000** | **0.000±0.000** | **0.000±0.000** | **0.000±0.000** | **0.000±0.000** |
| n200 | 0.050±0.002 | **0.000±0.000** | **0.000±0.000** | **0.000±0.000** | **0.000±0.000** | **0.000±0.000** | 0.004±0.000 | **0.000±0.000** |
| n300 | 0.042±0.006 | **0.000±0.000** | **0.000±0.000** | **0.000±0.000** | **0.000±0.000** | **0.000±0.000** | 0.000±0.000 | **0.000±0.000** |
| Avg. | 0.024 | **0.000** | 0.031 | 0.019 | 0.015 | 0.029 | 0.017 | **0.000** |

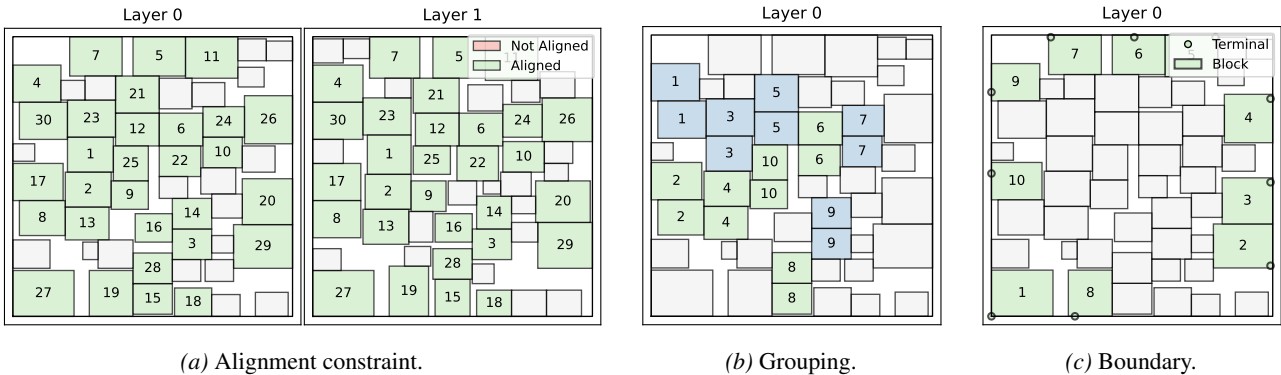

*(a) Alignment constraint.*        *(b) Grouping.*        *(c) Boundary.*

*Figure 8.* 3D floorplan demonstration with complex hardware design rules on Task 3.

## M. More Experiments about Zero-Shot Inference

We conduct additional experiments on zero-shot inference (mentioned in Sec. 5.3 in the main paper). In Table 22, we first train the policy network on circuit n100 to solve Task 1, which involves boundary and inter-die block alignment constraints. We then evaluate its performance on other circuits without further fine-tuning. Similarly, in Table 23, we apply the same procedure to assess zero-shot performance on Task 3, which includes boundary, alignment, and grouping constraints. The *ratio* in the tables represents the metric ratio between zero-shot inference and training on the corresponding circuit. These results demonstrate that our approach exhibits strong zero-shot transferability.

When evaluating the zero-shot inference capability of our policy, we draw inspiration from the commonly used test-time scaling techniques (Snell et al., 2024) in large language model (LLM) inference. Specifically, we retain the top $K$ checkpoints of policy network during the training process. For each checkpoint, we conduct $M$ independent inference attempts in parallel during testing to perform the 3D floorplanning task. Note that the total number of generated samples is $N = K \times M$. We adopt a best-of-$N$ (BoN) strategy, whereby the best candidate among the floorplan solutions without overlap is selected as the final output for zero-shot inference and is subsequently used for evaluation. For instance, the reward associated with the terminal state can be employed as the selection criterion.

*Table 22.* Zero-shot transferability evaluation on Task 1 (training on circuit n100).

| Metric/Circuit | | ami33 | ami49 | n10 | n30 | n50 | n200 | n300 |
|---|---|---|---|---|---|---|---|---|
| Blk-Tml Distance (↓) | value | 0.000 | 0.000 | 0.000 | 0.000 | 0.000 | 0.000 | 0.000 |
| | ratio | 1.000 | 1.000 | 1.000 | 1.000 | 1.000 | 1.000 | 1.000 |
| Alignment (↑) | value | 0.939 | 0.882 | 0.970 | 0.910 | 0.956 | 0.856 | 0.896 |
| | ratio | 0.992 | 0.958 | 1.008 | 0.988 | 1.051 | 0.931 | 0.984 |
| HPWL (↓) | value | 68,246 | 1,017,943 | 31,746 | 94,490 | 114,570 | 358,650 | 497,979 |
| | ratio | 1.168 | 1.278 | 1.008 | 0.997 | 0.966 | 1.081 | 1.087 |
| Overlap (↓) | value | 0.000 | 0.000 | 0.000 | 0.000 | 0.000 | 0.000 | 0.000 |
| | ratio | 1.000 | 1.000 | 1.000 | 1.000 | 1.000 | 1.000 | 1.000 |

*Table 23.* Zero-shot transferability evaluation on Task 3 (training on circuit n100).

| Metric/Circuit | | ami33 | ami49 | n10 | n30 | n50 | n200 | n300 |
|---|---|---|---|---|---|---|---|---|
| Blk-Blk Adj. Length (↑) | value | 0.159 | 0.153 | 0.124 | 0.177 | 0.203 | 0.109 | 0.095 |
| | ratio | 0.757 | 0.890 | 0.905 | 0.927 | 0.940 | 0.916 | 0.941 |
| Blk-Tml Distance (↓) | value | 0.000 | 0.000 | 0.000 | 0.000 | 0.000 | 0.000 | 0.000 |
| | ratio | 1.000 | 1.000 | 1.000 | 1.000 | 1.000 | 1.000 | 1.000 |
| Alignment (↑) | value | 0.944 | 0.961 | 0.913 | 0.903 | 0.977 | 0.904 | 0.867 |
| | ratio | 1.007 | 1.026 | 0.971 | 0.988 | 0.994 | 0.946 | 0.950 |
| HPWL (↓) | value | 72,231 | 873,835 | 35,676 | 96,653 | 121,878 | 362,026 | 477,390 |
| | ratio | 1.148 | 0.985 | 1.110 | 1.017 | 1.014 | 1.048 | 1.032 |
| Overlap (↓) | value | 0.000 | 0.000 | 0.000 | 0.000 | 0.000 | 0.000 | 0.000 |
| | ratio | 1.000 | 1.000 | 1.000 | 1.000 | 1.000 | 1.000 | 1.000 |

## N. More Experiments about Fine-Tune

We conduct additional experiments to test the fine-tune transferability (mentioned in Sec. 5.3 in the main paper) of our approach. Firstly, we train the policy network on circuit n100 to solve corresponding task. Based on these pre-trained weights, we further fine-tune it on other unseen circuits. To test the scalability of our approach, we concentrate on fine-tuning on larger circuits, including n200 and n300, which contain more blocks, nets and terminals than n100. Corresponding fine-tune results are shown in 1) Fig. 9, on circuit n200 with Task 1; 2) Fig. 11, on circuit n300 with Task 1; 3) Fig. 12, on circuit n300 with Task 2. Through the fine-tuning technique, better or similar performance can be achieved, conserving substantial training resources.

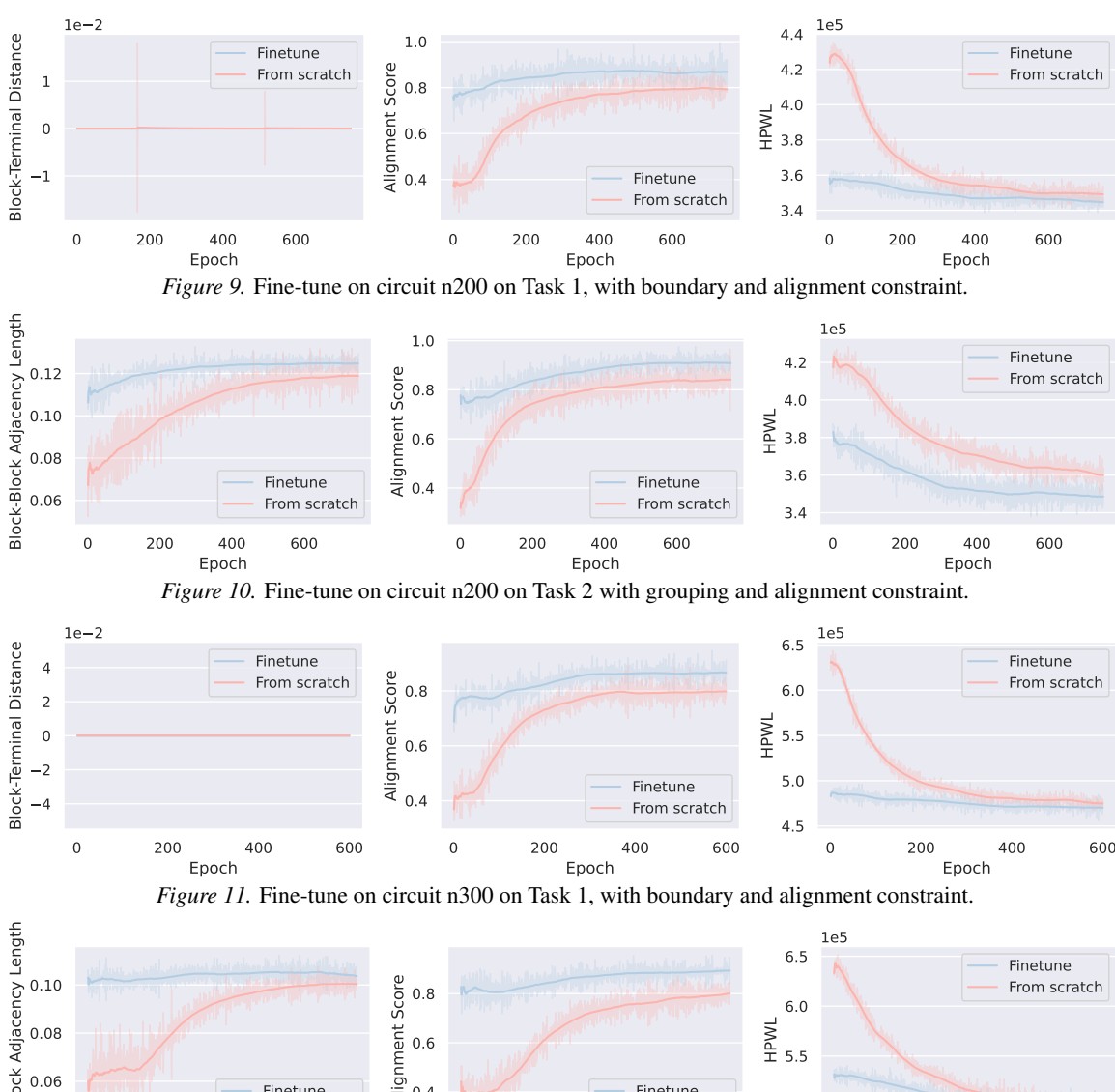

*Figure 9.* Fine-tune on circuit n200 on Task 1, with boundary and alignment constraint.

*Figure 10.* Fine-tune on circuit n200 on Task 2 with grouping and alignment constraint.

*Figure 11.* Fine-tune on circuit n300 on Task 1, with boundary and alignment constraint.

*Figure 12.* Fine-tune on circuit n300 on Task 2, with grouping and alignment constraint.

## O. Experiments on Constraint Adherence with Thresholds

For a more comprehensive evaluation, we introduce thresholds to determine whether a design rule is violated and count the number of blocks satisfying each rule. This quantitative measure provides a clearer understanding of constraint adherence.

Separate thresholds are assigned for constraints (a), (b), and (c). A block–block or block–terminal pair is considered to satisfy a design rule only if it meets the specified threshold. Otherwise, it is marked as unsatisfied. The threshold settings for the three design rules are as follows:

1. For block–terminal distance, the threshold is set to $0$. A block–terminal pair with a distance less than this threshold is considered valid.

2. For block–block adjacency length, the threshold is set to half the minimum common edge length of the two adjacent modules, i.e., $0.5 \times \min\{l_1, l_2\}$. Only block–block pairs with an adjacency length greater than this threshold are considered valid.

3. For inter-die block alignment, the threshold is set to half the minimum area of the two modules, i.e., $0.5 \times \min\{a_1, a_2\}$. Only block–block pairs with an alignment area exceeding this threshold are considered valid.

Based on these threshold, we judge whether a block satisfy the corresponding rule, and summarize the number of blocks satisfying the rule. Results are shown in Table 24, 25, 26.

*Table 24.* Number of blocks satisfying the design constraint (a) on Task 1.

| C/M | B*-3D-SA | GraphPlace | DeepPlace | MaskPlace | WireMask-BBO | FlexPlanner | RulePlanner |
|---|---|---|---|---|---|---|---|
| ami33 | 1/5 | 2/5 | 0/5 | 0/5 | 0/5 | 2/5 | **5/5** |
| ami49 | 1/5 | 0/5 | 0/5 | 2/5 | 0/5 | 0/5 | **5/5** |
| n10 | 1/2 | 0/2 | 0/2 | **2/2** | 0/2 | 0/2 | **2/2** |
| n30 | 0/5 | 0/5 | 0/5 | 1/5 | 0/5 | 1/5 | **5/5** |
| n50 | 1/5 | 0/5 | 0/5 | 0/5 | 0/5 | 1/5 | **5/5** |
| n100 | 0/10 | 0/10 | 0/10 | 3/10 | 0/10 | 2/10 | **10/10** |
| n200 | 0/10 | 0/10 | 1/10 | 1/10 | 0/10 | 0/10 | **10/10** |
| n300 | 0/10 | 0/10 | 0/10 | 0/10 | 0/10 | 1/10 | **10/10** |

*Table 25.* Number of blocks satisfying the design constraint (b) on Task 2.

| C/M | B*-3D-SA | GraphPlace | DeepPlace | MaskPlace | WireMask-BBO | FlexPlanner | RulePlanner |
|---|---|---|---|---|---|---|---|
| ami33 | 1/10 | 3/10 | 2/10 | 7/10 | 4/10 | 3/10 | **10/10** |
| ami49 | 3/10 | 0/10 | 2/10 | 2/10 | 4/10 | 2/10 | **10/10** |
| n10 | 2/2 | 0/4 | 3/4 | **4/4** | **4/4** | 3/4 | **4/4** |
| n30 | 1/10 | 1/10 | 1/10 | 1/10 | 3/10 | 3/10 | **10/10** |
| n50 | 0/10 | 0/10 | 0/10 | 0/10 | 1/10 | 2/10 | **10/10** |
| n100 | 2/20 | 1/20 | 1/20 | 4/20 | 2/20 | 2/20 | **20/20** |
| n200 | 0/20 | 1/20 | 2/20 | 4/20 | 0/20 | 4/20 | **20/20** |
| n300 | 0/20 | 2/20 | 0/20 | 4/20 | 0/20 | 1/20 | **20/20** |

*Table 26.* Number of blocks satisfying the design constraint (c) on Task 3.

| C/M | B*-3D-SA | GraphPlace | DeepPlace | MaskPlace | WireMask-BBO | FlexPlanner | RulePlanner |
|---|---|---|---|---|---|---|---|
| ami33 | 16/20 | 0/20 | 2/20 | 1/20 | 0/20 | 12/20 | **20/20** |
| ami49 | 9/20 | 1/20 | 1/20 | 0/20 | 1/20 | 19/20 | **20/20** |
| n10 | 9/10 | 3/10 | 2/10 | 7/10 | 2/10 | 9/10 | **10/10** |
| n30 | 17/20 | 3/20 | 4/20 | 10/20 | 0/20 | 17/20 | **20/20** |
| n50 | 14/20 | 1/20 | 1/20 | 2/20 | 4/20 | 19/20 | **20/20** |
| n100 | 25/60 | 1/60 | 1/60 | 3/60 | 1/60 | 51/60 | **60/60** |
| n200 | 10/60 | 2/60 | 0/60 | 3/60 | 0/60 | 51/60 | **60/60** |
| n300 | 0/60 | 1/60 | 0/60 | 2/60 | 1/60 | 44/60 | **60/60** |

