# OpenReview forum: "RulePlanner: All-in-One Reinforcement Learner for Unifying Design Rules in 3D Floorplanning"
_ICML.cc/2026/Conference — ICML 2026 regular_

### Official Review · Reviewer_hsGj · 2026-03-09

**Soundness:** 4
**Presentation:** 4
**Significance:** 3
**Originality:** 2
**Overall Recommendation:** 4
**Confidence:** 4

**Summary:**

This paper proposes a deep reinforcement learning framework named RulePlanner to handle multiple complex hardware design rules in 3D IC floorplanning. By transforming the design rules into matrix inputs and quantifying them to reward signals for optimization, RulePlanner is able to handle various constraints. For hard rules that can not be violated, RulePlanner incorporates direct constraints on the action space to filter invalid positions. Experimental results on public benchmarks demonstrate that RulePlanner significantly outperforms traditional analytical, heuristic, and RL-based baselines in constraint satisfaction while maintaining robust zero-shot generalization and transferability to unseen circuits.

**Compliance With Llm Reviewing Policy:**

Affirmed.

**Final Justification:**

My main concerns have been addressed. I keep my original positive score.

**Key Questions For Authors:**

(1) In MaskPlace, the connectivity between macros is implicitly encoded via WireMask, bypassing the need for explicit graph inputs. Could the authors provide an ablation study comparing the current approach against a vanilla CNN-based architecture? Additionally, have the authors compared the performance of GNN-based versus Transformer-based encoders for capturing graph features?

(2) Given that the aspect ratio head predicts the configuration for the next module, does this imply that the initial module in the placement sequence must be a hard module? How to determine the placement order? Furthermore, as the input feature of the network, what is the specific methodology for determining the initial width, height, and area of modules?

**Limitations:**

Yes

**Strengths And Weaknesses:**

Strengths:

(1) The main contribution of this paper is to introduce the important design rules into 3D floorplan and demonstrate that those design rules can be represented as matrices, and that quantification can transform them into reward signals to facilitate policy learning.

(2) The paper is technically sound and features comprehensive empirical analysis, including main results, generalization results, and ablation studies.

(3) The paper is clearly written and well-structured.

Weaknesses:

(1) While this work builds on solid foundations, it is important to clarify the provenance of several key technical components. Specifically, the use of matrix-based state features and dense reward signals can be traced back to MaskPlace [1], an RL-based macro placer that first introduced these designs. More recently, yet 3-4 months prior to this submission, EXPlace [2] adapted and extended these techniques to systematically incorporate diverse expert knowledge into the placement process. Given that EXPlace targets a closely related placement problem and, notably, pursues a conceptually similar direction of encoding domain expertise and design rules within the learning framework, I believe that an appropriate discussion should be included in the paper.

(2) The paper does not provide a training time comparison with other RL-based methods. Given the increase in optimization objectives, the convergence time is a critical factor that remains unexplored.

(3) The HPWL comparison in Table 12 is not inappropriate. Since the HPWL scales vary across different benchmarks, calculating the mean for HPWL is inappropriate. Instead, the average rank can be used as a more robust alternative.


[1] MaskPlace: Fast Chip Placement via Reinforced Visual Representation Learning, NeurIPS2022.

[2] Expertise can be helpful for reinforcement learning-based macro placement, ICLR2026.

---

> ### Author Rebuttal · Authors · 2026-03-30
>
> ### 1. Prior Work Acknowledgment & Distinguished Contributions
>
> We thank the reviewer for highlighting MaskPlace and EXPlace. We agree that the paper should acknowledge these connections more clearly. Matrix-/mask-based representations and dense step-wise rewards are not introduced by our work: MaskPlace introduced visual mask-based placement, and EXPlace further encoded expert knowledge with specialized masks. RulePlanner builds on this line of work, but our claim is not ingredient-level novelty. Rather, our contribution is a 3D representation-feasibility-policy interface that jointly handles cross-die coupling, hard feasibility, and soft-block geometry.
>
> **RulePlanner's Key Distinctions:**
>
> * **3D vs. 2D:** MaskPlace/EXPlace target 2D placement, while RulePlanner targets 3D floorplanning with cross-die dependencies, requiring new representations such as the inter-die alignment mask.
> * **Constraint handling:** Prior works mainly use masks for expert guidance. RulePlanner uses rule matrices as features, rewards, and an action-feasibility mask to enforce all four rule types as hard constraints. In the main text, overlap and block-terminal distance are reported as violation magnitudes, while alignment and adjacency are reported as continuous scores; the Appendix reports binarized 0/1 satisfaction for the latter two.
> * **Hybrid action space:** RulePlanner jointly predicts discrete placement and continuous soft-block aspect ratios, which are absent in fixed-shape 2D formulations.
>
> Accordingly, we will revise the paper to state the novelty more precisely: not mask-based RL representations per se, but a unified representation-feasibility-policy interface for 3D floorplanning with heterogeneous rules and infeasibility filtering.
>
>
>
> ### 2. Training Time Comparison
>
> We agree that convergence cost is important. On the representative N100 circuit, under the same hardware/software environment and the same 1000-epoch training schedule, the per-epoch runtime is:
>
> | Method | Runtime / epoch |
> | :--- | :--- |
> | MaskPlace | 16 s |
> | DeepPlace | 17 s |
> | GraphPlace | 17 s |
> | FlexPlanner | 18 s |
> | RulePlanner | 19 s |
>
> RulePlanner is only 1-3 s slower per epoch despite modeling three additional rule matrices. Under this matched 1000-epoch setting, all methods have entered late-stage convergence after about epoch 900 on N100, so the runtime comparison is informative for practical training cost. We will clarify in the revision that this result measures runtime overhead under matched convergence behavior on N100.
>
>
> ### 3. HPWL Metric Normalization
>
> We agree that arithmetic means across benchmarks with very different HPWL scales are inappropriate. We therefore use average rank for cross-benchmark comparison:
>
>
> | Circuit | Analytical | B*-3D-SA | GraphPlace | DeepPlace | MaskPlace | WireMask-BBO | FlexPlanner | RulePlanner |
> | :--- | :--- | :--- | :--- | :--- | :--- | :--- | :--- | :--- |
> | ami33 | 4 | 6 | 8 | 3 | 7 | 2 | 5 | 1 |
> | ami49 | 8 | 3 | 5 | 6 | 7 | 4 | 2 | 1 |
> | n10 | 2 | 7 | 8 | 3 | 5 | 1 | 4 | 6 |
> | n30 | 1 | 5 | 3 | 4 | 8 | 2 | 6 | 7 |
> | n50 | 1 | 8 | 3 | 4 | 7 | 2 | 5 | 6 |
> | n100 | 1 | 8 | 7 | 3 | 5 | 2 | 4 | 6 |
> | n200 | 1 | 7 | 8 | 5 | 6 | 2 | 4 | 3 |
> | n300 | 1 | 7 | 6 | 5 | 8 | 2 | 4 | 3 |
> | **Average Rank** | **2.375** | **6.375** | **6.000** | **4.125** | **6.625** | **2.125** | **4.250** | **4.125** |
>
> RulePlanner achieves an average rank of 4.125, tying DeepPlace, while also maintaining hard-constraint satisfaction; the main text reports overlap and block-terminal distance directly, and the Appendix gives binarized 0/1 satisfaction for alignment and adjacency.
>
> ### 4. CNN vs. GNN vs. Transformer Encoders
>
> Ablation on N100:
>
> | Architecture | Avg. HPWL | Block-Term Dist. | Adjacency | Alignment |
> | :--- | :--- | :--- | :--- | :--- |
> | CNN | 201,458 | 0.000 | 0.140 | 0.901 |
> | GNN | 199,674 | 0.000 | 0.143 | 0.913 |
> | Transformer | **197,302** | **0.000** | **0.147** | **0.924** |
>
> * **CNN:** simple baseline, but weaker on long-range dependencies.
> * **GNN:** improves connectivity modeling, reducing HPWL by ~0.9% and raising Alignment to 0.913.
> * **Transformer:** best overall, with ~2.0% HPWL reduction vs. CNN and Alignment 0.924.
>
>
> ### 5. Initial Module Configuration & Aspect Ratio Prediction
>
> **a) Virtual block:** We prepend a virtual block (w=h=0, no netlist connections). Its step predicts the aspect ratio for the first real block, so all real modules can be treated uniformly as soft blocks.
>
> **b) Placement order:** Rule-constrained blocks (boundary, grouping, inter-die alignment) are placed first; the remaining blocks are sorted by descending area.
>
> **c) Initial dimensions:** Fixed area comes from the benchmark specification. Initial width/height come from the circuit specification. For soft blocks, each predicted aspect ratio updates width/height while preserving area, and the updated geometry is fed back into the next step.

---

> > ### Author Rebuttal · Reviewer_hsGj · 2026-04-02
> >
> > Thank you for your thorough response. Most of my concerns have been well addressed. I will maintain my current positive score.

---

> > > ### Author Response · Authors · 2026-04-05
> > >
> > > Thank you for the follow-up and for the constructive assessment.
> > >
> > > In the revision, we will incorporate the suggested clarifications on related work, runtime comparison, HPWL normalization, and the architecture / initialization details.
> > >
> > > Thank you again for helping us improve the paper.

---

### Official Review · Reviewer_6uij · 2026-03-11

**Soundness:** 3
**Presentation:** 3
**Significance:** 2
**Originality:** 2
**Overall Recommendation:** 4
**Confidence:** 3

**Summary:**

This paper studies 3D floorplanning and proposes RL-based RulePlanner. The main idea is to encode different design rules with rule-specific matrix representations, constrain the action space by masking invalid placements, and incorporate rule satisfaction into the reward. The reported results show strong performance on rule-related metrics such as boundary satisfaction, grouping, and cross-die alignment, together with ablation studies and zero-shot / fine-tuning transfer results.

**Compliance With Llm Reviewing Policy:**

Affirmed.

**Final Justification:**

The rebuttal has addressed my concerns, I will raise my score to positive.

**Key Questions For Authors:**

1. What is the main general-ML novelty here relative to prior work on constrained RL?
2. In the zero-shot experiments, how much of the reported gain comes from the best-of-N checkpoint-and-sampling procedure rather than from the policy itself?

**Limitations:**

Yes.

**Strengths And Weaknesses:**

Strengths:

1. The method is clear and the rule representations are reasonable.
2. The results are strong and design rule satisfaction
3. Rich ablations and experiments.

Weaknesses:

1. Is this paper a strong fit for ICML? The main novelty lies in rule-engineered matrix representations for EDA. The methodological contribution feels closer to effective problem-specific engineering.
2. The paper does not sufficiently disentangle the contribution of the learning algorithm from the contribution of hand-engineered constraints. The ablation study mainly examines whether these masks are used as features and constraints. However, the paper does not compare against a simpler non-RL or greedy constrained baseline using the same rule matrices / availability mask.
3. Since the optimization pipeline itself is not fully circuit-agnostic, this weakens the claim that the learned policy generalizes cleanly across circuits.

---

> ### Author Rebuttal · Authors · 2026-03-30
>
> ### 1. Methodological Novelty & ICML Fit
>
> Our claim is not that action masking or matrix features are individually new. Our contribution is a constrained RL interface that combines rule compilation, feasibility filtering, and policy learning for 3D floorplanning. It converts heterogeneous geometric/topological rules into a shared representation that one policy can handle jointly while preserving solution quality.
>
> ### 2. Isolating Learning Algorithm from Hand-Engineered Constraints
>
> To isolate policy learning from hand-engineered constraints, we add a non-RL baseline, **WireMask-BBO-Constraint**, that uses the same rule matrices and availability masks but replaces the learned policy with constrained search. The table reports averages over the full benchmark suite:
>
> | Metric | WireMask-BBO | +Constraint | RulePlanner |
> |--------|--------------|-------------|-------------|
> | Alignment Score | 0.065 | 0.179 | **0.938** |
> | Block-Block Adjacency | 0.054 | 0.083 | **0.162** |
> | HPWL | 303,693 | 314,480 | **275,334** |
> | Overlap | 0.029 | 0.040 | **0.000** |
> | Block-Terminal Distance | 0.672 | 0.052 | **0.000** |
>
> **Analysis:**
> 1. **Constraint Effect:** Adding constraints to WireMask-BBO improves Alignment by 2.7x and reduces Block-Terminal Distance by 92.6%, confirming that the rule matrices and masks are useful.
>
> 2. **Learning Value:** Relative to this constrained non-RL baseline, RulePlanner still achieves 5.2x higher Alignment Score (0.938 vs. 0.179), 1.95x better Adjacency, 12.4% lower HPWL, and zero overlap / terminal-distance violations. This shows that the gains are not explained by hand-engineered constraints alone.
>
> 3. **Takeaway:** The improvement comes from interaction: the rule interface defines a structured feasible region, and the learned policy exploits it at high quality.
>
> ### 3. Circuit-Agnostic Representations & Generalization
>
> We agree that the full optimization pipeline is not entirely circuit-agnostic, since reward normalization is circuit-dependent. Our claim is narrower: the transferable component is the representation-policy interface. The design-rule matrices translate diverse circuit constraints into a shared 2D spatial representation consumed by the same policy across circuits.
>
> The evidence is: (1) **Zero-Shot Transferability:** a policy trained on n100 produces zero-shot placements on substantially different circuits (ami33 to n300) without parameter updates, while maintaining more than 90% of trained performance on the reported metrics; and (2) **Accelerated Fine-Tuning:** pre-training on smaller circuits speeds convergence on larger unseen circuits relative to training from scratch. We will revise the paper text to focus this claim on representation/policy transferability rather than the entire pipeline.
>
> ### 4. General-ML Novelty in Constrained RL
>
> From a constrained-RL perspective, our main contribution is to compile heterogeneous, non-differentiable spatial rules into a shared interface for representation, feasibility control, and optimization. The transferable part is this interface-level formulation rather than any single ingredient.
>
> 1. **Representation-Driven Rule Compilation:** We map complex, non-differentiable spatial rules into dense matrices used jointly for state representation, reward shaping, and dynamic action filtering. This is the main ML-facing contribution we claim.
>
> 2. **Unified Handling of Heterogeneous Constraints:** Our formulation separates rule representation, discrete action filtering, and continuous spatial projection, allowing overlapping geometric and topological constraints to be handled within one policy interface. We believe this is the most transferable aspect beyond EDA.
>
> ### 5. Best-of-N Sampling Analysis
>
> To separate the effect of best-of-N selection from the effect of the policy itself, we also evaluate a simpler protocol: use only the last checkpoint trained on n100 and run a single zero-shot inference on each target circuit, without checkpoint or sample selection. The numbers below are the ratio of this single-run protocol to the BoN result, averaged over the transferred circuits.
>
> * **Single-run performance remains strong:** Relative to BoN, the single-run protocol reaches 1.0477x HPWL, 0.9821x Alignment, and 0.9549x Block-Block Adjacency, while keeping Overlap and Block-Terminal Distance at 0.
> * **BoN still provides gains:** This matches the common observation in LLMs and other stochastic generative settings that best-of-N sampling improves final quality by exploiting output diversity. We therefore do not argue that BoN is unnecessary; it provides an additional improvement on top of an already strong zero-shot policy.
> * **Efficiency tradeoff:** Running several inference samples on a target circuit is far cheaper than fine-tuning or retraining on that circuit. Thus, BoN is a practical inference-time improvement, while the single-run result shows that the transferred policy is already effective without target-specific adaptation.

---

> > ### Author Rebuttal · Reviewer_6uij · 2026-04-05
> >
> > Thank you for your rebuttal, I will raise my score to positive.

---

> > > ### Author Response · Authors · 2026-04-05
> > >
> > > Thank you for the follow-up, for considering our clarification satisfactory, and for updating your assessment positively.
> > >
> > > In the revision, we will further sharpen the paper's scope and contribution claims, and clarify the constrained non-RL baseline, the narrower representation/policy transfer claim, and the role of best-of-N sampling in zero-shot evaluation.
> > >
> > > We appreciate the careful reading and constructive feedback.

---

### Official Review · Reviewer_1tAf · 2026-03-12

**Soundness:** 2
**Presentation:** 3
**Significance:** 3
**Originality:** 4
**Overall Recommendation:** 4
**Confidence:** 2

**Summary:**

This paper proposes RulePlanner, a reinforcement-learning framework for 3D floorplanning under multiple hardware design rules. The method represents rules with spatial matrices, uses availability masks to eliminate invalid actions, and combines rule-aware rewards with hybrid PPO. The author combines meaningful design rule definition, not just HPWL, but handles boundary, grouping, alignment, non-overlap, outline, and shape constraints within a unified framework. Evaluation results show clear advantages over several analytical, heuristic, and prior learning-based baselines.

**Compliance With Llm Reviewing Policy:**

Affirmed.

**Key Questions For Authors:**

1. The paper argues that satisfying complex rules is more important than achieving the lowest HPWL. Can the authors quantify this trade-off more explicitly, for example, by summarizing how large the HPWL degradation is when RulePlanner achieves substantially better rule satisfaction? Is there a way in your method to trade off HPWL with rule satisfaction?

2.  Thermal considerations are more important in 3D floorplanning. Do the authors have any results related to thermal? What are the main obstacles to extending the current framework to downstream concerns such as thermal constraints and routability?

3. It would strengthen the paper to include additional evidence on more industrially realistic designs and compare with an expert-tuned floorplan solution. Also, regarding the rule definitions in Section 2.2, can the authors further explain the source of these industrial design rules, including where these rules are drawn from and how they map to realistic 3D floorplanning settings?

**Limitations:**

yes

**Strengths And Weaknesses:**

This paper works on 3D floorplanning that combines multiple realistic rules in one RL framework. The proposed rule matrices provide structured guidance with availability masks that enforce hard feasibility. The main evaluation results are also strong. On the reported primary metrics, RulePlanner achieves block-terminal distance 0.000 on Task 1 and block-block adjacency length 0.223 on Task 2, versus a baseline of 0.082 for adjacency. This paper also shows zero-shot transferable performance to other circuits, which is hard for RL-based methods. However, its HPWL performance drops in a number of evaluation circuits. The main weaknesses are in evaluation realism and justification. The strongest claims rely on author-defined task constructions built on top of public MCNC/GSRC circuits, rather than on a benchmark that natively targets 3D floorplanning with these full industrial constraints. It would be nice to see the performance comparison with an industrial expert-tuned floorplanning solution to validate the applicability of this method.

---

> ### Author Rebuttal · Authors · 2026-03-30
>
> ### 1. Scalability and Practical Relevance
>
> We agree that additional evidence on realistic large-scale designs would strengthen the paper. In this work, we evaluate RulePlanner on substantially larger instances than those in FloorSet: the largest circuit in FloorSet contains **120 blocks**, whereas our dataset includes **n300** with **300 blocks**. This more than doubles the problem size under the same types of multi-rule constraints.
>
> We also evaluated RulePlanner on the largest circuits in our benchmark suite, Adaptec 2 and 3 (567 and 724 blocks, Task 3), and observed consistent improvements over WireMask-BBO:
>
> | Circuit | Method | Block-Block Adjacency | Alignment | HPWL | Overlap | Block-Terminal Distance |
> | :--- | :--- | :--- | :--- | :--- | :--- | :--- |
> | Adaptec 2 | WireMask-BBO | 0.049 | 0.099 | 4,136,779 | 0.0 | 0.000 |
> | Adaptec 2 | RulePlanner | **0.106** | **0.835** | **3,975,862** | 0.0 | 0.000 |
> | Adaptec 3 | WireMask-BBO | 0.027 | 0.101 | 8,190,146 | 0.0 | 0.000 |
> | Adaptec 3 | RulePlanner | **0.092** | **0.806** | **7,837,461** | 0.0 | 0.000 |
>
> These results support that RulePlanner remains effective as scale increases: it improves the rule-related metrics by a large margin and, on these large instances, also preserves competitive or better HPWL. We emphasize, however, that these open-benchmark results are supporting evidence for scalability rather than a replacement for proprietary industrial evaluation or expert-crafted floorplans. They show that the method does not break down as circuit size and constraint complexity grow.
>
> ### 2. HPWL Trade-off and Controllability
>
> We will clarify this trade-off more explicitly in the revision. Compared with unconstrained baselines, enforcing complex design rules leads to only **~3.23% average HPWL degradation** across benchmarks.
>
> Our framework can trade off HPWL and rule satisfaction by adjusting the corresponding reward coefficients. All four rule types are treated as hard constraints; in the main paper, Alignment and Adjacency are reported as continuous quality metrics, and in the Appendix we additionally report their binarized 0/1 satisfaction rates. Our default setting prioritizes rule satisfaction first and then optimizes secondary objectives such as HPWL, because layouts with severe rule violations would still require substantial post-processing even if their wirelength were lower.
>
> We will make this point clearer in the paper and add a more explicit summary of the HPWL-rule trade-off.
>
> ### 3. Thermal Constraints and Extensibility
>
> We appreciate this question. At present, we do **not** have thermal experimental results in the current submission, and we will state this limitation explicitly.
>
> That said, the framework itself can be extended to thermal objectives in a straightforward way: thermal maps or power-density features can be added to the policy input, and thermal metrics can be incorporated into the reward. The main obstacle is computational rather than architectural. RL training requires a very large number of environment interactions, and invoking accurate thermal solvers or routability estimators at each step would make training prohibitively slow. A practical next step is to use lightweight surrogate models for thermal or congestion estimation during training, which we view as future work rather than current experimental coverage.
>
> ### 4. Rule Source and Expert-Tuned Comparison
>
> The rule definitions in Section 2.2 are drawn from **FloorSet (Mallappa et al., ICCAD 2024)**, which was introduced to bridge academic benchmarks and practical SoC floorplanning constraints. In our setting, these rules correspond to realistic 3D design requirements: grouping constraints model tightly coupled modules or power domains, boundary constraints capture edge-aligned interface blocks, and inter-die alignment models vertically connected modules across dies.
>
> We agree that comparison against an expert-tuned industrial floorplan would further strengthen the paper. However, for open benchmarks with these combined 3D constraints, such expert-crafted reference solutions are not publicly available or easily reproducible. In this submission, we therefore compare against strong automated baselines, including WireMask-BBO, FlexPlanner, and MaskPlace. Across these baselines, the key distinction is that RulePlanner achieves substantially higher rule satisfaction while maintaining competitive wirelength. We will clarify both the origin of the rules and the limitation of currently available open benchmarks in the revised manuscript.

---

> > ### Author Rebuttal · Reviewer_1tAf · 2026-04-03
> >
> > Thanks for the detailed response. My concerns are partially resolved but I still maintain a positive score.

---

> > > ### Author Response · Authors · 2026-04-03
> > >
> > > Thank you for the follow-up and for the constructive assessment. We agree that validation beyond open benchmarks, especially on more industrially realistic designs and downstream objectives such as thermal/routability, would further strengthen the paper.
> > >
> > > At the same time, we would like to clarify the intended scope of this work. Our primary focus is on geometric/positional design rules in 3D floorplanning, including adjacency, boundary, alignment, and block-terminal relations. We view this problem as practically important yet still comparatively underexplored in the open literature, particularly under combined multi-rule settings. Our goal is therefore not to claim that we already cover the full range of industrial objectives, but rather to address a meaningful and insufficiently studied layer of floorplanning constraints that current open baselines do not yet handle well.
> > >
> > > In the revision, we will make this scope clearer, further clarify the source and practical motivation of these rules, and add a more explicit summary of the HPWL-versus-rule-satisfaction trade-off. We appreciate the suggestion and will present industrial validation and thermal integration as important future directions.

---

### Official Review · Reviewer_jmmw · 2026-03-13

**Soundness:** 3
**Presentation:** 3
**Significance:** 3
**Originality:** 3
**Overall Recommendation:** 4
**Confidence:** 4

**Summary:**

This paper addresses the problem of handling multiple real-world hardware design rules simultaneously in 3D floorplanning, and proposes RulePlanner, a unified deep RL framework for this setting. The core idea is to represent different design rules within a common pipeline using dedicated rule matrices, action-space constraints that directly filter invalid placements, and quantitative rule-specific metrics that are incorporated into reward computation. RulePlanner introduces new matrix representations such as adjacent terminal masks and adjacent block masks, combines them with alignment and position masks to construct an availability mask over valid actions, and uses an actor-critic RL framework with a hybrid action space to jointly decide block positions and aspect ratios. Experiments on public benchmarks show that the proposed method performs strongly against prior analytical, heuristic, and RL-based baselines, while also demonstrating promising generalization and transferability.

**Compliance With Llm Reviewing Policy:**

Affirmed.

**Final Justification:**

My concerns are addressed. I maintained a positive score.

**Key Questions For Authors:**

Please refer to the numbered items in the Weaknesses section above.

**Limitations:**

Yes.

**Strengths And Weaknesses:**

**Strengths:**
1. The problem of handling multiple real-world hardware design rules at the same time in 3D floorplanning is important.
2. The main idea of the paper is clear. It presents a unified RL framework that brings together rule-specific representations, action-space constraints, and quantitative rule metrics in a single pipeline.
3. The experimental results are strong overall and suggest that the proposed method compares favorably with prior analytical, heuristic, and RL-based baselines.

**Weaknesses:**
1. The paper introduces many components at once, including new matrix representations, action masking, rule-specific metrics, and the overall RL framework. As a result, it is hard to tell which parts are driving the improvements. A more detailed ablation study would make the paper stronger.
2. The related work could be broadened a bit. In addition to prior floorplanning methods, it would be helpful to briefly discuss other stages of the chip design pipeline, such as logic synthesis [1][2], as well as PPA-aware placement studies [3]. This would help better position the paper within the broader landscape of ML for chip design.
3. Although the paper emphasizes extensibility, the current evaluation is still based on a fixed set of rules and benchmark settings. It remains unclear how easily the framework would transfer to substantially different rule sets or more complex industrial scenarios.
4. The paper mainly highlights strong rule-satisfaction results, but the discussion of trade-offs between satisfying complex rules and optimizing conventional objectives such as wirelength could be more thorough.
5. The empirical results are promising, but the evaluation is still limited to benchmark circuits of moderate scale. It would be useful to better understand how the approach scales to larger and more diverse real-world designs.

---

> ### Author Rebuttal · Authors · 2026-03-30
>
> ### 1. Ablation Study & Component Contributions
> Our ablation study (Sec. 5.4, Table 6) isolates the components:
>
> * **Rule matrices and action masking:** Row `Feat.` removes the rule matrices from the policy input, and row `Constr.` removes the feasibility mask from the action space. Both cause clear drops in Table 6, showing that representation and masking are important.
>
> * **Policy learning vs. constraint engineering:** We added a constrained non-RL baseline that uses the same rule matrices and masks but replaces the learned policy with constrained search; these numbers are averaged over the full benchmark suite. Constraints alone improve several rule-related metrics, but RulePlanner still achieves much better alignment/adjacency and lower HPWL, showing that the main gain comes from combining the rule interface with a learned policy that exploits the feasible region it defines.
>
> * **Backbone / graph processing:** On N100, a CNN gives 201,458 HPWL / 0.901 Alignment, a GNN gives 199,674 / 0.913, and the Transformer performs best with **197,302 HPWL** and **0.924 Alignment**. We will add these ablations to the revision to clarify each component's contribution.
>
> * **Rule-specific metrics:** These are part of the task definition rather than optional modules. In the revision, we will make this distinction explicit: ablations isolate representation, masking, and policy/backbone choices, while the rule metrics define the optimization target.
>
> ### 2. Expanded Related Work
> We thank the reviewer for this suggestion. Due to page limits, we focused the submission on prior floorplanning methods. In the revision, we will broaden Related Work to include other chip-design stages, including logic synthesis [1][2] and PPA-aware placement [3], to better position our work within ML for chip design.
>
> ### 3. Extensibility to Complex Industrial Scenarios
> * The extensible part of RulePlanner is the representation-policy interface: a new rule is compiled into a matrix, a feasibility condition, and a reward term, while the policy architecture and training pipeline remain unchanged.
> * Our experiments instantiate this interface on **seven practical constraints**, whereas prior methods typically address only one or two specialized rules. Appendix E shows two additional industrial scenarios.
> * We will revise the text to make the claim more precise: we do not claim full industrial coverage, only that new heterogeneous geometric/topological rules can be added through the same interface instead of redesigning the optimizer.
>
> ### 4. Trade-offs: Rule Satisfaction vs. Wirelength (HPWL)
> We agree that this trade-off should be discussed more explicitly. Empirically, compared with unconstrained baselines, enforcing the complex design rules causes only **~3.23% average HPWL degradation** across benchmarks. Our framework can trade off HPWL and rule satisfaction by adjusting reward coefficients. All four rule types are hard constraints; in the main text, Alignment and Adjacency are continuous quality metrics, and the Appendix reports binarized 0/1 satisfaction rates. Our default setting prioritizes rule satisfaction first because layouts with severe rule violations would still require substantial post-processing even if their wirelength were lower. RulePlanner achieves competitive wirelength optimization: on the large-circuit results below, it improves rule-related metrics by a wide margin while achieving better HPWL than WireMask-BBO. We will make this trade-off more explicit in the revision.
>
> ### 5. Scalability to Larger Real-World Designs
> We further evaluated the method on larger benchmarks. The largest circuit in FloorSet contains **120 blocks**, while our dataset includes **n300** with **300 blocks** under the same multi-rule constraints. We also evaluated RulePlanner on Adaptec 2 and 3 for the most challenging Task 3. These contain 567 and 724 blocks, respectively. RulePlanner outperforms prior methods on rule metrics and achieves strong HPWL:
>
> | Adaptec 2 | Adjacency Length | Alignment | HPWL | Overlap | Block-Terminal Distance |
> | :--- | :--- | :--- | :--- | :--- | :--- |
> | WireMask-BBO | 0.049 | 0.099 | 4,136,779 | 0.0 | 0.000 |
> | RulePlanner | **0.106** | **0.835** | **3,975,862** | 0.0 | 0.000 |
>
> | Adaptec 3 | Adjacency Length | Alignment | HPWL | Overlap | Block-Terminal Distance |
> | :--- | :--- | :--- | :--- | :--- | :--- |
> | WireMask-BBO | 0.027 | 0.101 | 8,190,146 | 0.0 | 0.000 |
> | RulePlanner | **0.092** | **0.806** | **7,837,461** | 0.0 | 0.000 |
>
> These results suggest that RulePlanner remains effective as circuit size increases: it maintains strong rule satisfaction and preserves competitive or better HPWL on substantially larger instances. We agree that open benchmarks still do not fully cover the diversity of proprietary industrial designs; these results are therefore supporting evidence for scalability rather than a substitute for proprietary industrial validation, and we will state this limitation explicitly in the revision.

---

> > ### Author Rebuttal · Reviewer_jmmw · 2026-04-05
> >
> > Thanks for the detailed response. My concerns are addressed. I maintained a positive score. In addition, due to an oversight, I did not include the references when I submitted my review, so I am adding them here. I would be glad to see the authors include a discussion of these related works in the revised paper.
> > [1] A Graph Enhanced Symbolic Discovery Framework for Efficient Logic Optimization. ICLR 2025.
> > [2] Evolving Graph Structured Programs for Circuit Generation with Large Language Models. ICLR 2026.
> > [3] LaMPlace: Learning to optimize cross-stage metrics in macro placement. ICLR 2025.

---

> > > ### Author Response · Authors · 2026-04-05
> > >
> > > Thank you for the follow-up and for indicating that our rebuttal addressed your concerns. We also especially appreciate your adding the related references that were omitted in the original review.
> > >
> > > In the revision, we will incorporate these works and use them to better position RulePlanner within the broader ML-for-chip-design literature, while keeping our claim focused on the unified representation-feasibility-policy interface for multi-rule 3D floorplanning.
> > >
> > > Thank you again for the constructive feedback throughout the review process.

---

### Decision · Program_Chairs · 2026-04-30

**Decision:**

Accept (regular)

**Comment:**

This paper proposes RulePlanner, a unified deep reinforcement learning framework for 3D floorplanning under multiple realistic hardware design rules.
Most reviewers agreed on the importance of the problem, the clarity of the proposed method, the strength of the experimental results, and the overall quality of the presentation. The authors also made substantial efforts during the rebuttal phase and addressed most of the reviewers' concerns. During the reviewer-author discussion phase, the reviews overall trended toward acceptance. Overall, this paper offers a valuable contribution to the community. Therefore, I recommend accepting this paper.